# PREFERENCE-BASED PROCESS REWARD MODEL FOR ROBUST MATHEMATICAL REASONING

## ABSTRACT

Process reward models (PRMs) have emerged as a promising approach to guide LLMs by providing step-wise supervision, but traditional methods often rely on heuristic search strategies like Monte Carlo Tree Search (MCTS), which introduce bias and limit generalization. In this work, we propose a reinforcement learning framework guided by a Preference-Based Process Reward Model (PPRM). We first employ MCTS to estimate and select chosen and rejected rollouts, thereby constructing a high-quality step-level dataset. Our PPRM is trained on Bradley-Terry loss function, which mitigates the bias introduced by the heuristic search strategies of MCTS by leveraging preference-based learning and offers a more robust and theoretically grounded approach to reward modeling. To enable effective RL training with PPRM, we enhance Group Relative Policy Optimization (GRPO) by introducing a robust advantage estimator that better captures the structure of preference-based process reward model. Experimental results on Process-Bench and best-of-n strategy demonstrate that our approach achieves 2-3% improvement in intermediate step accuracy compared to existing methods for complex reasoning processes, thereby improving the reasoning accuracy of the policy model across several key reasoning benchmarks.

## 1 INTRODUCTION

Large language models (LLMs) have demonstrated impressive capabilities in mathematical reasoning (Yang et al., 2024)(Guo et al., 2025)(Grattafiori et al., 2024), solving complex problems by decomposing them into logical steps (Yao et al., 2023). However, they still face critical challenges, including calculation errors, flawed logical reasoning, and even the generation of fabricated or hallucinated intermediate steps. These issues undermine the reliability of LLMs in precise domains like mathematics, where accuracy and consistency are essential. Reinforcement learning (RL) has garnered significant attention (Ouyang et al., 2022)(Touvron et al., 2023). However, challenges remain in scaling these methods efficiently due to substantial computational requirements and the need for careful reward design to prevent shortcut learning (Cao et al., 2024)(Chan et al., 2024).

In the field of mathematical reasoning, reward models are typically categorized into two main types: the outcome reward model (ORM) and the process reward model (PRM). Specifically, the ORM cannot identify or rectify errors in intermediate steps, leading to potential suboptimal preformance (Lightman et al., 2023) where correct answers are derived from incorrect reasoning. Process Reward Model (PRM) offers a promising solution by providing step-wise reinforcement learning feedback. Existing work has shown consistent results that PRMs outperform ORMs in best-of-N sampling (Snell et al., 2024) and RL (Setlur et al., 2024).

**Limitations of PRM**. As highlighted in studies such as DeepSeek R1 (Guo et al., 2025), accurately determining the correctness of intermediate steps remains a challenging task. PRMs often struggle to provide reliable evaluations of intermediate results, ultimately affecting the accuracy and generalization of the final outcome.

**(i). Issues with Annotation**. A major challenge in training PRMs lies in obtaining accurate step-level annotations. Lightman et al. (Lightman et al., 2023) demonstrated the effectiveness of using human expert annotators to label intermediate reasoning steps, ensuring high-quality supervision for PRM training. To address this, researchers have turned to automated annotation methods, with the Monte Carlo (MC) estimation approach being one of the most widely adopted. This method,

popularized by Wang et al. (Wang et al., 2024b) and Lu et al. (Lu et al., 2024), involves sampling multiple reasoning trajectories to empirically estimate the correctness probability of each step.

**(ii). Inadequacy of MCTS in Automated Annotation**. Although efficient and scalable, MC-based methods often rely on Monte Carlo Tree Search (MCTS), a heuristic-driven algorithm that introduces significant bias (Guan et al., 2025). MCTS prioritizes certain reasoning paths based on its exploration-exploitation strategy, which can lead to the reinforcement of suboptimal or unjustified steps (Zhang et al., 2025), compromising the generalization ability of the trained PRM. It significantly relies on the performance of the completion model, which may generate correct answers based on incorrect steps, or incorrect answers based on correct steps, introducing substantial noise and inaccuracy verification into step-wise correctness estimation.

In this work, we leverage preference learning to debias the Process Reward Model, proposing the Process Reward Preference-Based Model (PPRM). We theoretically demonstrate that PPRM ensures more stable and generalizable learning compared to MCTS-based rewards. Additionally, applying RL to PPRM requires modifications to Generalized Reinforcement Learning with Policy Optimization (GRPO), as standard GRPO struggles with the non-stationarity induced by preference-based rewards, as illustrated in Fig 1. Our enhanced algorithm enables efficient optimization, leading to more robust reasoning in LLMs. The contribution of our work is summarized as follows:

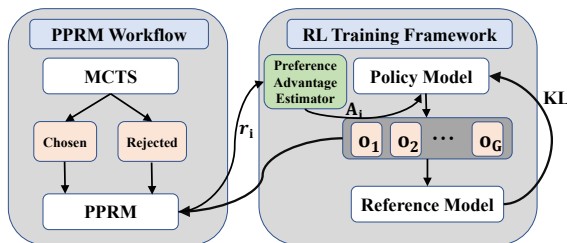

Figure 1: Illustration of PPRM framework. We apply MCTS to select chosen-rejected pairs accrording to $Q(s, r)$ to train PPRM and apply GRPO framework to PPRM with a preference advantage estimator.

- We introduce preference-based annotation into process reward modeling, reframing step-level supervision as pairwise comparisons between full reasoning trajectories. Through theoretical analysis grounded in the Bradley-Terry model (Bradley & Terry, 1952), we show that this formulation inherently mitigates bias and noise in automatic labeling—regardless of whether rollouts come from MCTS, LLM-as-a-judge, or other heuristic generators—by replacing error-prone absolute correctness estimates with robust relative judgments.

- We construct a high-quality dataset of expert-annotated trajectory preferences over mathematical reasoning steps, enabling the training of PPRM: a general-purpose, preference-based process reward model. PPRM consistently outperforms hard-label and soft-label PRMs across diverse annotation strategies (including MCTS and LLM-as-a-judge), demonstrating reduced sensitivity to the idiosyncrasies of any single rollout mechanism while achieving state-of-the-art step-wise accuracy.

- We propose a novel advantage estimator for GRPO that integrates seamlessly with the pairwise nature of preference signals. This estimator stabilizes policy optimization by aligning credit assignment with trajectory-level outcomes, leading to consistent gains across mathematical benchmarks—from elementary word problems to olympiad-level challenges—without relying on search-intensive or heuristic-dependent reward shaping.

## 2 RELATED WORKS

**Synthetic Data Generation.** Recent advances in training LLMs for mathematical reasoning have focused on generating high-quality process supervision data, with several key approaches emerging to address the trade-offs between annotation quality, scalability, and bias mitigation. Lightman (Lightman et al., 2023) pioneered expert-annotated step-level correctness labels to train PRMs, ensuring high fidelity but at significant cost, while Wang (Wang et al., 2024b) proposed scalable MC sampling to approximate step-wise correctness probabilities, trading off some precision for broader coverage. Luo (Luo et al., 2024) refined MC approaches with binary tree search, dynamically pruning incorrect reasoning paths during sampling to reduce noise. More recently, Zhang (Zhang et al.,

2025) introduced a hybrid approach combining LLM-based judger models with MC estimation, using the former to filter or reweight sampled trajectories. These methods collectively highlight the challenges in generating reliable process supervision data, which addressed by our work through the introduction of the BT model and robust advantage estimation, offering a more theoretically grounded and scalable solution for mathematical reasoning tasks.

**Preference Learning.** To address the challenge of reward bias, previous research has explored preference models for human alignment, particularly in cases where direct scoring is difficult. Preference learning allows for more flexible and interpretable reward modeling by comparing alternative outputs rather than assigning absolute scores (Ouyang et al., 2022)(Bai et al., 2022). This approach has proven effective in reducing bias in human feedback systems, making it a promising direction for improving reasoning in LLMs (Sun et al., 2025).

**RL Algorithm in Mathematical Reasoning.** Researchers have begun employing reinforcement learning (RL) for mathematical reasoning and introduced several sophisticated algorithms to enhance the reasoning capabilities of LLMs (Ouyang et al., 2022)(Snell et al., 2024). Proximal Policy Optimization (PPO) (Schulman et al., 2017)leverages clipped objective functions to ensure gradual policy updates while optimizing for both final answer correctness and intermediate reasoning quality. Building upon PPO, Reinforcement Learning from Online Oracle (RLOO) (Ahmadian et al., 2024) and Remax (Li et al., 2023) significantly reduces error propagation in multi-step derivations. Direct Preference Optimization (DPO) (Rafailov et al., 2023) and many of its variants (Azar et al., 2024)(Ethayarajh et al., 2024)(Chen et al., 2024) offer another innovative approach by directly optimizing policy outputs to align with human preferences without explicit reward modeling, simplifying the RL pipeline while maintaining strong performance. More recently, Group Relative Policy Optimization (GRPO) (Shao et al., 2024) has emerged as a promising alternative, employing group-wise comparisons of reasoning trajectories to prioritize logically consistent solutions over superficially correct but flawed answers.

## 3 PPRM Workflow

In this section, we introduces our methodology for enhancing multi-step reasoning in large language models through preference-based process reward model. We begin by employing Monte Carlo Tree Search (MCTS) to generate annotated reasoning trajectories. We then develop a PPRM that learns from relative comparisons between reasoning paths rather than absolute scoring, significantly reducing bias and improving generalization. We provide a theoretical comparison of two distinct annotation paradigms, Hard Estimation and Preference Estimation to evaluate their accuracy in reward modeling. The detailed proof can be found in Appendix A

### 3.1 Preference Pair Generating with Monte Carlo Method

Although Monte Carlo Tree Search (MCTS) is commonly used in automated annotation tasks, its reliance on heuristic strategies and stochastic sampling can lead to inconsistent or suboptimal results. This is particularly evident when dealing with tasks involving complex semantics or long-range dependencies, where MCTS often fails to deliver satisfactory performance, thereby limiting both the efficiency and quality of automated annotation.

Motivated by these challenges, we propose the preference annotation to construct high-quality problem-solving data pairs for training the process reward model in a preference-based format. Specifically, a "completer" policy is established that can take a question $q$ and a set of prefix solutions comprising the first $t$ steps $x_{1:t}$, ensuring the resulting data pairs are suitable for preference learning. We construct a Monte Carlo tree to represent the decision space, where each node corresponds to a state in the problem-solving process, and edges represent possible actions or steps. For each problem, multiple completions are sampled and organized into this tree structure, allowing us to evaluate and compare different reasoning trajectories.

To identify the most informative data pairs, we define a scoring mechanism based on the Q-value of each rollout. The Q-value balances the quality of the solution, as estimated by the Monte Carlo method, with its complexity, ensuring that the selected pairs are both high-quality and concise. Specifically, the probability of selecting a chosen rollout $r_{chosen}$ and a reject rollout $r_{reject}$ is cal-

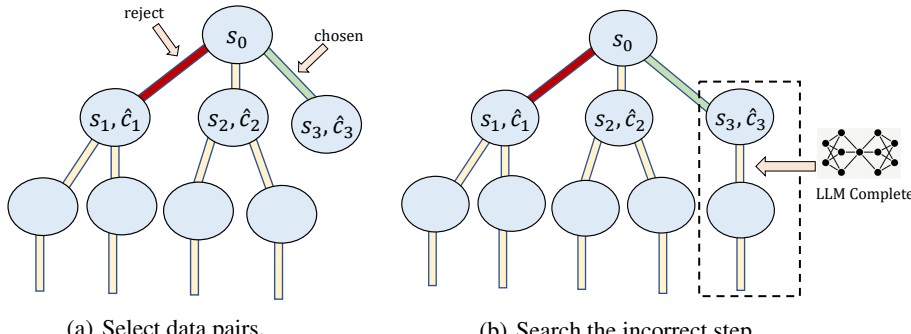

(a) Select data pairs.       (b) Search the incorrect step.

Figure 2: Illustration of preference pair generating with Monte Carlo method. (a) The workflow contains generating multiple completions for each problem, constructing an MC tree to evaluate these completions and assessing the rollouts using MC estimation. (b) The selection formula $Q(s, r)$ is applied to identify the optimal choice-reject pairs, which are subsequently compiled into a structured dataset.

culated using the following formulas:

$$Q_{\text{chosen}}(s, r) = \alpha^{1-MC(s)} \cdot \beta^{\text{len}(r)}, \quad Q_{\text{reject}}(s, r) = \alpha^{MC(s)} \cdot \beta^{\text{len}(r)}. \quad (1)$$

Here, $\alpha$ is a hyperparameter that adjusts the weight of the Monte Carlo estimation, $MC(s)$ represents the Monte Carlo estimation score for the state $s$, and $\beta$ accounts for the weight of the length of the rollout $r$. The term $\alpha^{1-MC(s)}$ ensures that higher-quality rollouts (as indicated by a higher $MC(s)$) are more likely to be chosen, while $\alpha^{MC(s)}$ prioritizes lower-quality rollouts for rejection. The term $\beta^{\text{len}(r)}$ penalizes overly complex solutions, favoring concise reasoning paths. Notably, the trajectory sampling method for step-level data collection demands significantly higher computational resources (FLOPs) compared to training ORMs.

As shown in Figure(2), the workflow begins by generating multiple completions for each problem, followed by the construction of an MC tree to evaluate and organize these completions. The rollouts are then assessed using the combined approach of MC estimation and implicit Q function. The selection formula $Q(s, r)$ is applied to identify the optimal choice-reject pairs, which are subsequently compiled into a structured dataset. This methodology ensures the creation of a dataset that captures diverse and accurate problem-solving processes, maintaining a balance between complexity and correctness for downstream tasks.

To further clarify the robustness of our method on various strategies, we compare two distinct approaches for sampling rollouts: Monte Carlo estimation and LLM-as-a-Judge. For the MC estimation approach, we train PRMs on chosen-rejected pairs selected by a scoring mechanism based on the Q-value of each rollout mentioned in Section 3.1. We use the same training and testing split as described in Lightman et al. (2023), which consists of 12K training examples and a subset with 500 holdout representative problems from the original 5K testing examples. We conduct the data generation process for training the PPRM utilizing Qwen2.5-Math-7B-Instruct as the completer model on the dataset. For the LLM-as-a-judge approach, we use the same query and completer and randomly sample our trajectories from the generated $k = 8$ rollouts. We employ Qwen2.5-7B-Instruct to verify the correctness of each step in the responses and implement the prompt template for verification follows Zhang et al. (2025), which is shown in Appendix B.2. We then construct chosen-reject pairs by randomly pairing rollouts labeled as correct and incorrect by the judge.

### 3.2 THE FORMULATION OF ANNOTATION

**Hard MC Estimation.** Most process-surpvised reward models (PRMs) are traditionally trained under the next token prediction framework (Zhang et al., 2024b), which aims to predict the likelihood of the next token in a sequence. PRM ($\mathcal{P} \times \mathcal{S} \rightarrow \mathbb{R}^+$) assigns a score $\hat{r}$ to each reasoning step of $s$,

which is usually trained with cross enropy loss function:

$$L_{\text{CE}} = \sum_{i=1}^{K} y_{p,s_i} \log \hat{r}_{s_i} + (1 - y_{p,s_i}) \log(1 - \hat{r}_{s_i}). \tag{2}$$

where $K$ is the number of reasoning steps in $s$, $\hat{r}_{s_i}$ is the output of PRM with given $s_i$ and $y_{s_i}$ is the ground truth label of the $i$-th step of $s$. Unlike common data annotations, the hard MC-estimated annotation $y(p, s_i, A)$ at the $i$-th step is actually a function of the ratio of correct rollouts to total rollouts from the $i$-th step.

$$y(p, s_i) = \begin{cases} 1, & c_i > \lambda, \\ 0, & \text{else.} \end{cases} \tag{3}$$

Here $c_i = c(p, s_i)$ denotes the ratio of correct rollouts to total rollouts from the $i$-th step and $\lambda$ represents the threshold for distinguishing between positive and negative labels based on the MC estimation.

However, the estimated ratio $\hat{c}_i = c_i + b(p, s_i, A)$ introduces bias which is a random variable depends on the annotator $A$. Therefore, the MC estimation needs to satisfy the following condition: $(\hat{c}(p, s_i, A) - \lambda)(c(p, s_i) - \lambda) > 0$. In other words, the labels estimated via MCTS must preserve the same ordering relationship as the threshold criterion.

We can map this ordering into a binary value through a increasing function $h_{\text{hard}}(c_i) = \mathbb{I}(c_i > \lambda)$. The label for $h_{\text{hard}}$ contains noise follows a Bernoulli distribution $\eta \sim \text{Bernoulli}(p_{\text{hard}})$ where $p_{\text{hard}}$ is given by

$$p_{\text{hard}} = p(\{b(p, s_i, A) : (c_i - \lambda)^2 < (\lambda - c_i) \cdot b(p, s_i, A)\}) = 1 - \xi_{\text{hard}}(\Delta c_i). \tag{4}$$

where $\Delta c_i = c_i - \lambda$ represents the difference between $c_i$ and threshold, and the approximating true scores $\xi_{\text{hard}}(\Delta c_i)$ is a function of $\Delta c_i$.

Thus the model actually trains on the noisy training dataset $\mathcal{D}_{\text{hard}} = \{(p_n, s_{p_n}, \hat{h}_n), n = 1, \cdots, N\}$ where the ordering label $y_n$ has noise $\eta$ defined by Eq.(4). The loss function in Eq.(2) often suffers from high variance, particularly in generative models, where the quality of the generated outputs is heavily dependent on the quality of the annotations used during training. This limitation can lead to suboptimal performance, especially in tasks requiring precise and contextually accurate outputs.

**Preference MC Estimation.** We don't require the reward model to predict probabilities of step-level label accurately, but rather to provide a reliable signal for ranking a group of LLM outputs at inference. An alternative approach is to employ the Bradley-Terry (BT) model, which is particularly well-suited for learning process reward from pairwise comparisons. In this framework, we select chosen-reject pairs $(p, s_1)$ and $(p, s_2)$ from the dataset $\mathcal{D} = \{(p_n, s_{p_n}, \hat{c}_n), n = 1, \cdots, N\}$ generated from MATH dataset in Section 3.1, where both pairs share the same prompt $p$ but differ in their responses $s$.

Specifically, the loss function for the BT model is defined as follows:

$$\mathcal{L}_{\text{BT}} = \mathbb{E}\left[\mathbb{I}_{h=1}\sigma(\hat{r}_{\text{BT}}(p, s_1) - \hat{r}_{\text{BT}}(p, s_2)) + \mathbb{I}_{h=-1}(1 - \sigma(\hat{r}_{\text{BT}}(p, s_1) - \hat{r}_{\text{BT}}(p, s_2)))\right]. \tag{5}$$

Here, $\mathbb{E}$ denotes the expectation over the sampled pairs, $h_i = \mathbb{I}(c(p, s_1^i) > c(p, s_2^i))$ is the ground truth ordering, $\hat{r}_{\text{BT}}$ represents the output of reward model with input $(p, s)$ and $\sigma$ is the sigmoid function. The loss function encourages the model to assign higher rewards to preferred responses and lower rewards to rejected ones, thereby learning the underlying preference structure.

We can explain by treating preference pairs as ordered data, where the ground truth preference is defined as $\hat{h}_i = \mathbb{I}(\hat{c}(p, s_1^i, A) > \hat{c}(p, s_2^i, A)) \in \{0, 1\}$. However, the estimated ratio $\hat{c}_i = c_i + b(p, s_i, A)$ introduces bias, leading to noisy labels $\hat{h}_i = h_i + \eta$, where the noise $\eta$ occurs with the probability $p_{\text{pref}}$ given by:

$$p_{\text{pref}} = p(\{\Delta b(p, s_1^i, s_2^i, A) : \Delta c_i < -\Delta b(p, s_1^i, s_2^i, A)\}) = 1 - \xi_{\text{pref}}(\Delta c_i). \tag{6}$$

where $\Delta b(p, s_1^i, s_2^i, A) = b(p, s_1^i, A) - b(p, s_2^i, A)$ represents the difference between the bias in $\hat{c}(p, s_1^i, A)$ and $\hat{c}(p, s_2^i, A)$, $\Delta c_i = c(p, s_1^i) - c(p, s_2^i)$ represents the difference between the correctness level of $(p, s_1^i)$ and $(p, s_2^i)$, and $\xi_{\text{pref}}(\Delta c^i)$ is a strictly increasing function of $\Delta c^i$. This formulation captures the likelihood that the bias difference outweighs the true reward difference, which is critical for understanding the noise structure in the preference data.

### 3.3 RETHINKING PREFERENCE REWARD MODEL TRAINED ON MC ANNOTATIONS

Thus to ensure high-quality training data for the reward model, it is crucial to filter preference pairs with the largest ratio differences $\Delta c_i$ . Formally, we have

**Assumption 1.** *The data pair $(p, s_1^i)$, $(p, s_2^i)$ selected using the MCTS method satisfies $\hat{c}(p, s_1^i, A) > \lambda$ and $\hat{c}(p, s_2^i, A) \leq \lambda$.*

**Assumption 2.** *The distribution of $\hat{c}_{\mathrm{pref}}^i$ estimated in preference annotation is consistent with the distribution of $\hat{c}_{\mathrm{hard}}^i$ in hard MC-estimated annotation, i.e. $\hat{c}_i = \hat{c}_{\mathrm{pref}}^i = \hat{c}_{\mathrm{hard}}^i$.*

With those noisy annotations, we can consider the order consistency with true reward function:

**Proposition 1.** *Suppose that the expected agreement between the estimated preference $\hat{h}$ and the order model $\hat{H} = \mathbb{I}(\hat{r}_{\mathrm{BT}}(p, s_1) > \hat{r}_{\mathrm{BT}}(p, s_2))$ achieves up to $1 - \epsilon$ error for some $\epsilon$ and $\delta$, i.e.,*

$$\mathbb{E}_{p,s_1,s_2} \left[ \mathbb{I} \left[ \hat{h} = \hat{H} \right] \right] \geq 1 - \delta\epsilon, \tag{7}$$

*Then with probability at least $1 - \delta$, we can derive the following probabilistic guarantee for the correctness of the estimated preference:*

$$\mathbb{E}_{p,s_1,s_2} \left[ \mathbb{I} \left( \hat{H} \cdot [c(p, s_1^i) - c(p, s_2^i)] \geq 0 \right) \Big| \Delta c_i \right] \geq (1 - 2\epsilon) \cdot \xi(\Delta c_i) + \epsilon. \tag{8}$$

This result shows that the probability of the estimated preference $\hat{H}$ aligning with the true reward difference $c(x, y_1) - c(x, y_2)$ is lower-bounded by a function of $\xi(c_1, c_2)$, indicating high confidence in the preference correctness. The theoretical analysis follows closely the work of Sun et al. (Sun et al., 2025). Furthermore, we assume that the bias introduced by the MC estimation with the same annotator A can be offset, i.e.,

**Assumption 3.** *For the preference annotated data pair $(p, s_1^i, c_1^i), (p, s_2^i, c_2^i)$, the bias introduced by the MC estimation with the same annotator A can be offset, meaning that the distribution of $\Delta b = b_1(p, s_1^i, A) - b_2(p, s_2^i, A)$ is concentrated around $0$. We will assume the probability density function value of the random variable $\Delta b$ at $\Delta b < \hat{c}_1^i - \hat{c}_2^i$ is always greater than the PDF value at $\Delta b > \hat{c}_1^i - \hat{c}_2^i$, i.e.,*

$$p_{\Delta b}(u) > p_{\Delta b}(v), \quad \forall u < \hat{c}_1^i - \hat{c}_2^i, v > \hat{c}_1^i - \hat{c}_2^i. \tag{9}$$

We can then compare the accurate rate of the hard annotation and our preference annotation with the noisy ordering:

**Lemma 1.** *For noisy dataset $\mathcal{D}_{\mathrm{pref}} = \{(p_n, s_{p_n}^1, s_{p_n}^2, \hat{h}_n), n = 1, \cdots, N\}$ with selected data pairs follow Assumption 1 and dataset $\mathcal{D}_{\mathrm{hard}} = \{(p_n, s_{p_n}, \hat{h}_n), n = 1, \cdots, N\}$, the accuracy of model trained on $\mathcal{D}_{\mathrm{pref}}$ is higher than $\mathcal{D}_{\mathrm{hard}}$ across the whole dataset, i.e.*

$$\mathbb{E}_{\mathcal{D}_{\mathrm{pref}}} \left[ \mathbb{I} \left( \hat{H} \cdot [c(p, s_1^i) - c(p, s_2^i)] \geq 0 \right) \right] > \mathbb{E}_{\mathcal{D}_{\mathrm{hard}}} \left[ \mathbb{I} \left( \hat{H} \cdot [c(p, s^i) - \lambda] \geq 0 \right) \right]. \tag{10}$$

We find that the model trained on the noisy preference annotated dataset $\mathcal{D}_{\mathrm{pref}}$ achieves higher overall accuracy compared to the model trained on the hard annotated dataset $\mathcal{D}_{\mathrm{hard}}$. Preference-based training better captures the relative quality of solutions across the entire dataset. Under noisy labels, pairwise comparisons provide more informative learning signals than hard annotated labels.

## 4 THE FRAMEWORK OF RL TRAINING

We integrate Preference Process Models into the Generalized Reinforcement Learning with GRPO framework to enhance the training of mathematical reasoning models. We consider a reinforcement learning framework where we model the process a math agent solves problem $q$ as a Markov Decision Process (MDP), defined by the tuple $(\mathcal{S}, \mathcal{A}, p, r, \gamma)$. Here, $\mathcal{S}$ denotes the state space, $\mathcal{A}$ the action space, $p : \mathcal{S} \times \mathcal{S} \times \mathcal{A} \rightarrow [0, 1]$ the transition dynamics, $r : \mathcal{S} \times \mathcal{A} \rightarrow \mathbb{R}$ the reward function, and $\gamma \in [0, 1)$ the discount factor. The agent's behavior is governed by a policy $\pi_\theta(a|s)$, parameterized by $\theta$, which defines a distribution over actions given a state. For each problem $q$, trajectories

$\{o_i\}_{i=1}^G$ are generated under the old policy $\pi_{\theta_{old}}(O|q)$. We optimize a policy $\pi_\theta(a|s)$ by maximizing the expected discounted cumulative reward, formalized as:

$$J_{\text{GRPO}}(\theta) = \mathbb{E}_{q \sim P(Q), \{o_i\}_{i=1}^G \sim \pi_{\theta_{old}}(O|q)} \left[ \frac{1}{G} \sum_{i=1}^G \frac{1}{|o_i|} \sum_{t=1}^{|o_i|} \frac{\pi_\theta(o_{i,t}|q, o_{i,<t})}{\pi_{\theta_{old}}(o_{i,t}|q, o_{i,<t})} \hat{A}_{i,t} - \beta \mathbb{D}_{KL} \left[ \pi_\theta || \pi_{\text{ref}} \right] \right].$$

Here $G$ is the number of trajectories, and $o_i$ denotes the $i$-th trajectory. Each trajectory $o_i$ has a length $|o_i|$, and $o_{i,t}$ refers to the action taken at step $t$ in the $i$-th trajectory. The terms $\pi_\theta(o_{i,t}|q, o_{i,<t})$ and $\pi_{\theta_{old}}(o_{i,t}|q, o_{i,<t})$ denote the probabilities of selecting action $o_{i,t}$ under the new policy $\pi_\theta$ and the old policy $\pi_{\theta_{old}}$, respectively. A commonly used estimation method for the advantage function $\hat{A}_{i,t}$ is normalized rewards.

$$\hat{A}_{i,t} = \tilde{r}_t = \frac{r_{i,t} - \text{mean}(r_t)}{\text{std}(r_t)}. \tag{11}$$

Note that the advantage function is typically defined by $A(x,y) = Q(x,y) - V(x)$, which is not align with the normalized reward $\hat{A}_{i,t}$. Assuming the output of reward model $r_{i,t}$ contains bias $b(q, o_{i,<t})$, the estimation of the advantage function becomes dependent on the accuracy of the reward signal and suffers from high variance when the group size $G$ is limited. Therefore, it is needed to propose a more robust advantage estimation formula.

As suggested by the objective function of BT model, we employ the sigmoid function $\sigma$, which introduces a smoothing effect, further stabilizing the advantage estimates by mitigating the impact of outliers or extreme rewards. The output reward values of PPRM for a pair of actions $(x, y_1), (x, y_2)$ can estimate the probability of obtaining a higher return, i.e. $p(y_1 > y_2) = \sigma(r(x, y_1) - r(x, y_2))$. We denote the correctness of the final output of the policy model in state $(x, y)$ by $o_{x,y} \in \{0, 1\}$. Therefore, $Q(x, y) = E(o_{x,y}) = \bar{p}_{o_{x,y}}$ and the baseline $V(x) = \frac{1}{G} \sum_i Q(x, y_i)$. Our advantage estimator leverages the strengths of preference learning to reduce bias and improve the robustness of the reward signal. Specifically, the preference-based advantage estimator is formulated as follows:

$$\hat{A}_{i,t} = \frac{1}{G-1} \sum_{j \neq i} \sigma(r_{i,t} - r_{j,t}) - \frac{1}{G(G-1)} \sum_i \sum_{j \neq i} \sigma(r_{i,t} - r_{j,t}). \tag{12}$$

Here, $A_{i,t}$ represents the advantage of action $i$ at time step $t$, $r_{i,t}$ and $r_{j,t}$ denote the rewards for actions $i$ and $j$ respectively, and $\sigma$ is a sigmoid function that maps the reward difference to a preference probability. This formulation aggregates pairwise comparisons across all actions, ensuring that the advantage estimate reflects the relative quality of actions rather than their absolute rewards. By focusing on relative comparisons rather than absolute rewards, the estimator mitigates the high variance often associated with traditional advantage estimation methods, which leads to more stable and efficient policy updates during training.

## 5 EXPERIMENTS

This section presents our comprehensive experimental framework for enhancing mathematical reasoning in LLMs through PPRM training and RL training. First, we detail the PPRM training pipeline, where models learn to evaluate intermediate reasoning steps using filtered MATH dataset problems and Monte Carlo-based value estimation. We then examine RL training methodologies, which leverage PRM-generated signals to refine the LLM's reasoning trajectory.

### 5.1 PPRM TRAINING

**Data Generation.** We conduct the data generation process for training the PPRM utilizing Qwen2.5-Math-7B-Instruct as the completer model on the MATH dataset (Hendrycks et al.) , a well-established benchmark for mathematical reasoning tasks. For each state $s$ in the reasoning process, 16 rollouts are generated to explore a wide range of reasoning trajectories, ensuring that the PRM could learn from diverse reasoning strategies and a search limit of 50 was set for each problem. To reduce noise and focus on challenging and informative examples, we filter problems that were

Table 1: Performance comparison of different 7B reward models in PROCESSBENCH across GSM8K, MATH, OlympiadBench and Omni-MATH.

| Model | GSM8K | | MATH | | OlympiadBench | | Omni-MATH | |
|---|---|---|---|---|---|---|---|---|
| | acc | F1 | acc | F1 | acc | F1 | acc | F1 |
| Math-Shepherd-PRM-7B | 0.786 | 0.582 | 0.721 | 0.594 | 0.693 | 0.372 | 0.662 | 0.554 |
| Qwen2.5-Math-7B-Math-Shepherd | 0.785 | **0.585** | 0.715 | 0.588 | 0.691 | 0.413 | 0.674 | 0.546 |
| Math-PSA | 0.763 | 0.576 | 0.711 | 0.582 | 0.681 | 0.422 | 0.672 | 0.543 |
| ER-PRM | 0.771 | 0.0.562 | 0.707 | 0.523 | 0.685 | 0.434 | 0.667 | 0.532 |
| Skywork-PRM-7B | **0.795** | 0.533 | 0.722 | 0.583 | 0.697 | 0.486 | 0.684 | 0.576 |
| EurusPRM-Stage2 | 0.784 | 0.521 | 0.708 | 0.502 | 0.701 | 0.417 | 0.664 | 0.556 |
| PPRM-7B | 0.776 | 0.512 | **0.733** | **0.612** | **0.734** | **0.577** | **0.712** | **0.645** |

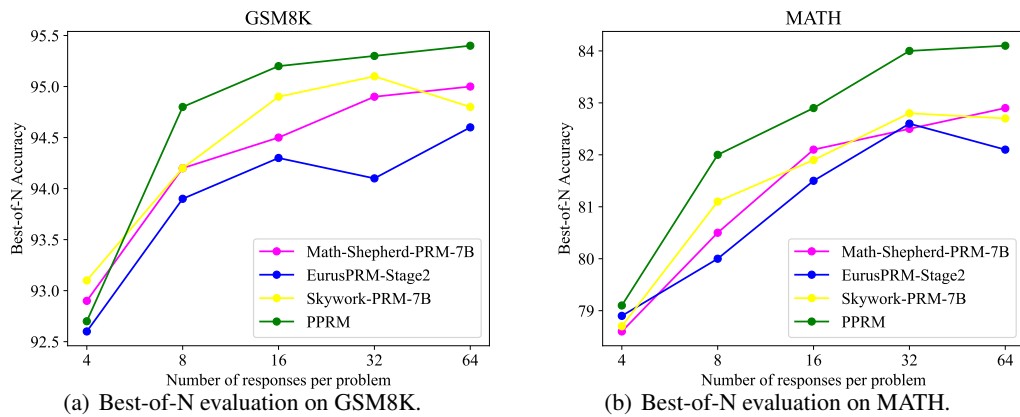

(a) Best-of-N evaluation on GSM8K.    (b) Best-of-N evaluation on MATH.

Figure 3: Best-of-N evaluation results on GSM8K and MATH datasets with Qwen2.5-Math-7B-Instruct as the generator.

either too simple or too difficult for the completer. To estimate the value $Q$ of each reasoning step, we use $\alpha = 0.5$ and $\beta = 0.9$ in Eq.(1) for each Monte Carlo estimation.

We follow the evaluation protocol established in Lightman et al. (2023). Our test set consists of a curated holdout subset of 500 representative problems drawn from the original MATH test set (which contains 5,000 problems). The training data for our reward models (including both the 12K human-annotated examples and our self-collected rollouts) excludes any problem overlapping with this 500-problem holdout set. The use of this standardized, community-accepted split which widely adopted in recent works minimizes the risk of leakage and enables fair comparison.

**Merics and Baseline.** We evaluate our PPRM on the ProcessBench performance and Best-of-N (BoN) strategy effectiveness, which evaluate the utilities of reward models in straightforwardly improving downstream task performance and the abilities of reward models to identify specific erroneous steps in reasoning processes. ProcessBench (Zheng et al., 2024) serves as our primary evaluation framework, comprehensively assessing models' ability to predict step-by-step reasoning correctness across challenging mathematical reasoning datasets: GSM8K (Cobbe et al., 2021) (elementary math problems), MATH (Hendrycks et al.) (advanced competition-level problems), OlympiadBench (He et al., 2024) (olympiad-style problems), and Omni-MATH (diverse mathematical reasoning tasks). This multi-dataset evaluation provides a robust measure of model performance across different difficulty levels and problem types. We also apply the Best-of-N strategy, which samples N reasoning paths and selects the one with the highest final-answer confidence. We compare our PPRM with Math-Shepherd-PRM-7B (Wang et al., 2024b), Qwen2.5-Math-7B-Math-Shepherd (Zhang et al., 2025), MATH-PSA (Wang et al., 2024a) which employs Omega PRM (Luo et al., 2024), Skywork-PRM-7B (Liu et al., 2024), ER-PRM Zhang et al. (2024a) and EurusPRM-Stage2 (Cui et al., 2025) trained using Implicit PRM (Yuan et al., 2024) which are trained on automated annotation data.

Table 2: Performance comparison of 7B reward models in PROCESSBENCH across GSM8K, MATH, OlympiadBench and Omni-MATH. The models are trained on the different data construction methods including MCTS and LLM-as-a-judge and labeling methods including PPRM, hard label and soft label.

|  | GSM8K | MATH | Olympiad Bench | Omni-MATH | Avg. |
|---|---|---|---|---|---|
| PPRM (MCTS) | 0.776 | 0.733 | 0.734 | 0.712 | 0.739 |
| PPRM (LLM-as-a-Judge) | 0.772 | 0.728 | 0.736 | 0.717 | 0.738 |
| Hard Label (MCTS) | 0.768 | 0.706 | 0.690 | 0.687 | 0.713 |
| Hard Label (LLM-as-a-Judge) | 0.770 | 0.709 | 0.682 | 0.678 | 0.710 |
| Soft Label (MCTS) | 0.765 | 0.696 | 0.674 | 0.670 | 0.701 |

**Evaluation Results.** We trained 7B-parameter PPRM, initialized with Qwen2.5-Math-7B-Instruct with the BT loss function on the dataset constructed above. We report the performance of preference process model on ProcessBench in Table 1. Our PPRM demonstrates superior overall performance on ProcessBench with the highest average accuracy and F1 scores across the four datasets, indicating its balanced precision-recall capabilities. Notably, PPRM excels in OlympiadBench and Omni-MATH, suggesting specialized strengths in olympiad-style challenges. These results underscore the importance of applying preference annotations in step-wise evaluation in refining LLM reasoning, with PPRM achieve better error identification balance and accuracy across datasets.

The results in the Table 2 provide performance comparison of 7B reward models in PROCESS-BENCH across GSM8K, MATH, OlympiadBench and Omni-MATH. Specifically,(1) when trained on the same underlying dataset—whether derived from MCTS or LLM-as-a-judge—the reward models using our PPRM consistently outperform those trained with hard labels or soft labels. This advantage is particularly pronounced on more complex benchmarks such as OlympiadBench and Omni-MATH, where fine-grained discrimination between subtly flawed and logically sound reasoning steps is critical. (2) Comparing data construction methods reveals a trade-off: LLM-as-a-judge yields superior generalization on challenging, out-of-distribution problems, while MCTS-based estimation performs slightly better on simpler, more standardized tasks like GSM8K.

The results best of N strategy on GSM8K and MATH with Qwen2.5-Math-Instruct-7B as the policy model in Figure 3 exhibit consistent performance improvements of PPRM with increasing sample sizes from 4 to 64. suggesting PPRM's architectural advantages in leveraging larger candidate pools through its robust preference learning framework. Especially on MATH, there is a significant gap of the accuracy between the two training methods. One hypothesis is that for challenging datasets like MATH, PPRM can deliver more robust reward signals with lower variance. The findings particularly emphasize PPRM's robust generalization, making it a promising approach for reliable mathematical reasoning.

## 5.2 RL TRAINING

We conduct RL training based on Qwen2.5-Math-1.5B and Qwen2.5-Math-7B. The training data consists of chain-of-thought format questions from the MATH dataset. For reward modeling, we compare our PPRM with Math-Shepherd, EurusPRM-Stage2, and MATH-PSA. For GRPO and RLOO implementation, we set the policy model learning rate to 1e-6 with a KL coefficient of 0.001. During exploration, we generate 8 outputs per question with a maximum sequence length of 1024 tokens. The training batch size is configured as 128 to balance memory constraints and training efficiency. For advantage estimator in GRPO, we employ the normalized estimator and our advantage estimator in Eq (12).

We repeat the experiment for 10 times and report the average score of the policy model initialized by Qwen2.5-Math-7B in Table (3) and Table (4). The performance of the policy model initialized by Qwen2.5-Math-1.5B can be found in Table (5) and Table (6) in Appendix B.1. The results of PPRM are demonstrated in the performance of various models on challenging datasets including GSM8K (Cobbe et al., 2021), AMC (Li et al., 2024), MATH (Hendrycks et al.), and Olympiad Bench. Our PPRM achieves the highest scores in AMC and MATH , while PPM+standard performs strongly in Olympiad Bench. Table (6) demonstrates GRPO with our improved preference-based advantage estimator shows the strongest performance, particularly excelling in MATH and AMC

Table 3: The performance of policy model initialized by Qwen2.5-Math-7B trained with PRMs on GRPO.

| | GSM8K | AMC | MATH | Olympiad Bench | AIME |
|---|---|---|---|---|---|
| ORM | $93.24 \pm 0.25$ | $38.84 \pm 0.55$ | $70.78 \pm 0.44$ | $49.87 \pm 0.83$ | $10.31 \pm 0.12$ |
| Math-Shepherd-PRM-7B | $95.22 \pm 0.11$ | $44.47 \pm 0.42$ | $74.03 \pm 0.27$ | $52.46 \pm 0.54$ | $16.71 \pm 0.26$ |
| Math-PSA | $94.02 \pm 0.07$ | $21.49 \pm 0.45$ | $73.88 \pm 0.29$ | $52.55 \pm 0.47$ | $13.33 \pm 0.21$ |
| Skywork-PRM-7B | $94.36 \pm 0.05$ | $45.73 \pm 0.47$ | $74.47 \pm 0.31$ | $53.04 \pm 0.19$ | $15.82 \pm 0.14$ |
| EurusPRM-Stage2 | $94.52 \pm 0.08$ | $44.49 \pm 0.64$ | $73.80 \pm 0.21$ | $51.15 \pm 0.15$ | $16.24 \pm 0.21$ |
| PPRM | $\mathbf{95.83 \pm 0.11}$ | $\mathbf{47.97 \pm 0.42}$ | $\mathbf{76.44 \pm 0.25}$ | $\mathbf{56.01 \pm 0.34}$ | $\mathbf{18.87 \pm 0.23}$ |

Table 4: The performance of policy model initialized by Qwen2.5-Math-7B trained with PRMs on RLOO and GRPO with various advantage estimators.

| | GSM8K | AMC | MATH | Olympiad Bench |
|---|---|---|---|---|
| RLOO | 95.4 | 48.3 | 76.8 | 54.5 |
| ReMax | 94.5 | 45.4 | 75.6 | 54.9 |
| GRPO | 95.8 | 47.9 | 76.3 | 55.2 |
| GRPO-P | **96.0** | **49.7** | **78.2** | **56.8** |

outperforming both standard GRPO and RLOO. The results highlight that while the baseline remains competitive in simpler tasks like GSM8K, PPRM combined with enhanced GRPO delivers more robust performance in complex reasoning scenarios.

## 6 DISCUSSION

We propose a reinforcement learning framework based on a Preference Process Reward Model (PPRM) to address the challenge in Reinforcement learning of obtaining precise step-level annotations for process reward models in multi-step reasoning tasks. Our enhanced algorithm enables efficient optimization, leading to more robust reasoning in LLMs. A primary area for improvement involves the computational demands of the MCTS process. Although MCTS is less expensive than extensive human annotation, its computational overhead remains substantial, potentially limiting the scalability of the approach to more complex or longer-horizon reasoning tasks. Our future work would prioritize exploring more efficient MCTS variants or alternative simulation-based methods.

## 7 ETHICS STATEMENT

Our paper complies with the ICLR Code of Ethics.

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

# A COMPARISON OF ORDER CONSISTENCY PROPERTIES BETWEEN HARD ANNOTATION AND PREFERENCE ANNOTATION.

## A.1 PROOF OF PROPOSITION 1

*Proof.* The idea of the proof is to first use Markov's inequality to bound probability that for a given distance $c_1 - c_2$ the preference model not well approximate the annotator and under the event that the preference model approximate the annotator well, we bound the total error combined by preference model and annotator.

By assumption we have the (marginal) error probability averaging over the dataset is

$$\mathbb{P}_{x,y_1,y_2,h}\left[1\left(\hat{H} \neq h\right)\right] = \mathbb{E}_{c_1,c_2}\left[\mathbb{P}\left(\hat{H} \neq h\big|c_1,c_2\right)\right] < \delta\epsilon. \tag{13}$$

by Markov's inequality,

$$\mathbb{P}_c\left(\mathbb{P}\left(\hat{H} \neq h\big|c_1,c_2\right) \geq \epsilon\right) \leq \frac{\delta\epsilon}{\epsilon} = \delta. \tag{14}$$

In the event $\{(c_1, c_2) : \mathbb{P}\left(\hat{H} \neq h \middle| c_1, c_2\right) < \epsilon\}$, with probability $1 - \delta$ we bound the error rate as function of $c_1, c_2$. Condition on $c_1, c_2$, we can discuss the order consistency of the learned model $\hat{H}_{\theta^*}$ with the oracle utility

- Correct Case: When the annotator is correct, the learned model agrees with the annotator with probability at least $1 - \epsilon$. Thus:

$$p_{\text{correct}} = p(\hat{H} = \hat{h}|\hat{h} = h) \geq (1 - \epsilon). \tag{15}$$

- Incorrect Case: When the annotator is incorrect, the learned model agrees with the annotator with probability at most $\epsilon$. Thus:

$$p_{\text{incorrect}} = p(\hat{H} \neq \hat{h}|\hat{h} \neq h) \leq \epsilon. \tag{16}$$

The order consistency of the learned model $\hat{H}_{\theta^*}$ with the oracle utility can be expressed as:

$$\mathbb{E}_{x,y_1,y_2 \sim \ell(x)} \left[ \mathbb{I}\left(\hat{H}(r(y_1, x) - r(y_2, x)) \geq 0\right) \middle| c_1, c_2\right] = p_{\text{correct}} \cdot p(\hat{h} = h) + p_{\text{incorrect}} \cdot p(\hat{h} \neq h)). \tag{17}$$

Substituting the bounds and simplifying, we have

$$\mathbb{E}_{x,y_1,y_2 \sim \ell(x)} \left[ \mathbb{I}\left(\hat{H}(r(y_1, x) - r(y_2, x)) \geq 0\right) \middle| c_1, c_2\right] \geq (1 - 2\epsilon) \cdot \xi(c_1, c_2) + \epsilon. \tag{18}$$

$$\square$$

### A.2 Proof of Lemma 1

*Proof.* We employ two different annotations, the hard annotation and the preference annotation to the same dataset $\mathcal{D} = \{(p_n, s_n, \hat{c}_n), n = 1, \cdots, N\}$ and obtain annotated dataset $\mathcal{D}_{\text{hard}} = \{(p_n, s_n, \hat{c}_n, \hat{h}_{\text{hard}}), n = 1, \cdots, N\}$ and $\mathcal{D}_{\text{pref}} = \{(p_n, s_n^1, s_n^2, \hat{c}_n^1, \hat{c}_n^2, \hat{h}_{\text{pref}}), n = 1, \cdots, \frac{N}{2}\}$. We will compare the accurate rate of the hard annotation and our preference annotation with the noisy ordering.

For hard estimation, the probability of noise is given by Eq.(4). Substituting into the Eq.(8), we have

$$\mathbb{E}_{p,s} \left[ \mathbb{I}\left(\hat{H} \cdot [c_i - \lambda] \geq 0\right) \middle| \Delta c_i\right] \geq (1 - 2\epsilon) \cdot p(\{b(p, s_i, A) : (c_i - \lambda)^2 < (\lambda - c_i) \cdot b(p, s_i, A)\}) + \epsilon. \tag{19}$$

The accuracy of model $\hat{H}_{\text{hard}}$ trained on $\mathcal{D}_{\text{hard}}$ across the dataset can be expressed by the sum of conditions when $c_i - \lambda > 0$ and when $c_i - \lambda < 0$:

$$\mathbb{E}_{\mathcal{D}_{\text{hard}}} \left[ \mathbb{I}\left(\hat{H} \cdot [c(p, s^i) - \lambda] \geq 0\right)\right] = \sum_{\hat{c}} \left[ \mathbb{I}\left(\hat{H} \cdot [\hat{c} - \lambda] \geq 0\right) \middle| \Delta c_i\right] \tag{20}$$

$$\geq \sum_{\hat{c} < \lambda} \left[(1 - 2\epsilon) \cdot p(\hat{c} - \lambda > b) + \epsilon\right] \tag{21}$$

$$+ \sum_{\hat{c} > \lambda} \left[(1 - 2\epsilon) \cdot p(\hat{c} - \lambda < b) + \epsilon\right]. \tag{22}$$

For preference estimation, the probability of noise is given by Eq.(6). Substituting into the Eq.(8), we have

$$\mathbb{E}_{p,s} \left[ \mathbb{I}\left(\hat{H} \cdot [c_i - \lambda] \geq 0\right) \middle| \Delta c_i\right] \geq (1 - 2\epsilon) \cdot p(\hat{c}(p, s_1^i) - \hat{c}(p, s_2^i) < b(p, s_1^i, A) - b(p, s_2^i, A)) + \epsilon. \tag{23}$$

The accuracy of model $\hat{H}_{\text{pref}}$ trained on $\mathcal{D}_{\text{pref}}$ across the dataset can be expressed by:

$$\mathbb{E}_{\mathcal{D}_{\text{pref}}} \left[ \mathbb{I}\left(\hat{H} \cdot [c_1 - c_2] \geq 0\right)\right] = \sum_{\hat{c}_1, \hat{c}_2} \mathbb{E}_{p, s_1, s_2} \left[ \mathbb{I}\left(\hat{H} \cdot [c_1 - c_2] \geq 0\right) \middle| \hat{c}_1 - \hat{c}_2\right] \tag{24}$$

$$\geq \sum_{\hat{c}_1, \hat{c}_2} \left[(1 - 2\epsilon) \cdot p(\hat{c}_1 - \hat{c}_2 > b_1 - b_2) + \epsilon\right]. \tag{25}$$

Note that the expectation in Eq.(20) and Eq.(24) are summed over the same dataset $\mathcal{D} = \{(p_n, s_n, \hat{c}_n), n = 1, \cdots, N\}$. Therefore,

$$\mathbb{E}_{\mathcal{D}_{\mathrm{pref}}}\left[\mathbb{I}\left(\hat{H} \cdot [c_1 - c_2] \geq 0\right)\right] - \mathbb{E}_{\mathcal{D}_{\mathrm{hard}}}\left[\mathbb{I}\left(\hat{H} \cdot [c(p, s^i) - \lambda] \geq 0\right)\right] \tag{26}$$

$$\geq (1 - 2\epsilon) \sum_{\hat{c}_1, \hat{c}_2} \left[p(\hat{c}_1 - \hat{c}_2 > b_1 - b_2) - \frac{1}{2}\left(p(\hat{c}_1 - \lambda > b_1) + p(\hat{c}_2 - \lambda < b_2)\right)\right]. \tag{27}$$

The ranges of correct ratio $c$ and $\hat{c}$ are both $(0, 1)$. Thus the range of bias $b = \hat{c} - c \in (\hat{c} - 1, \hat{c})$. Since $b_1$ and $b_2$ are independent, the probability $p(\hat{c}_1 - \hat{c}_2 > b_1 - b_2)$, $p(\hat{c}_1 - \lambda < b)$ and $p(\hat{c}_2 - \lambda < b)$ can be expressed in integral of $b_1$ and $b_2$ as:

$$P(\hat{c}_1 - \hat{c}_2 > b_1 - b_2) = \int_{\hat{c}_2 - 1}^{\hat{c}_2} \int_{\hat{c}_1 - 1}^{\hat{c}_1 - \hat{c}_2 + b_2} p(b_1)p(b_2)\, db_1\, db_2. \tag{28}$$

$$P(\hat{c}_1 - \lambda > b_1) = \int_{\hat{c}_1 - 1}^{\hat{c}_1 - \lambda} p(b_1)\, db_1 = \int_{\hat{c}_2 - 1}^{\hat{c}_2} \int_{\hat{c}_1 - 1}^{\hat{c}_1 - \lambda} p(b_1)p(b_2)\, db_1\, db_2. \tag{29}$$

$$P(\hat{c}_2 - \lambda < b_2) = \int_{\hat{c}_2 - \lambda}^{\hat{c}_1} p(b_2)\, db_2 = \int_{\hat{c}_1 - 1}^{\hat{c}_1} \int_{\hat{c}_2 - \lambda}^{\hat{c}_2} p(b_2)p(b_1)\, db_2, db_1. \tag{30}$$

Substituting the integrals and simplifying, we have

$$p(\hat{c}_1 - \hat{c}_2 > b_1 - b_2) - \frac{1}{2}(p(\hat{c}_1 - \lambda > b_1) + p(\hat{c}_2 - \lambda < b_2))$$

$$= \frac{1}{2}\left[\int_{\hat{c}_2 - 1}^{\hat{c}_2 - \lambda} \int_{\hat{c}_1 - 1}^{\hat{c}_1 - \hat{c}_2 + b_2} p(b_1)p(b_2)\, db_1\, db_2 - \int_{\hat{c}_2 - 1}^{\hat{c}_2 - \lambda} \int_{\hat{c}_1 - \hat{c}_2 + b_2}^{\hat{c}_1 - \lambda} p(b_1)p(b_2)\, db_1\, db_2\right]$$

$$+ \frac{1}{2}\left[\int_{\hat{c}_2 - \lambda}^{\hat{c}_2} \int_{\hat{c}_1 - \lambda}^{\hat{c}_1 - \hat{c}_2 + b_2} p(b_1)p(b_2)\, db_1\, db_2 - \int_{\hat{c}_2 - \lambda}^{\hat{c}_2} \int_{\hat{c}_1 - \hat{c}_2 + b_2}^{\hat{c}_1 - 1} p(b_1)p(b_2)\, db_1\, db_2\right].$$

Performing the variable substitution $u = \frac{b_1 + b_2}{2}$, $v = \frac{b_1 - b_2}{2}$. Note that the integration regions $\Omega_1 = \{(b_1, b_2) : b_1 - b_2 < \hat{c}_1 - \hat{c}_2, \hat{c}_2 - 1 < b_2 < \hat{c}_2 - \lambda\}$ and $\Omega_2 = \{(b_1, b_2) : b_1 - b_2 > \hat{c}_1 - \hat{c}_2, \hat{c}_2 - 1 < b_2 < \hat{c}_2 - \lambda\}$ are symmetric with respect to $v = \frac{\hat{c}_1 - \hat{c}_2}{2}$. Therefore, the first term of integral can be expressed by

$$\iint_{\Omega_1} p_u(u)p_v(v) - p_u(u)p_v(\hat{c}_1 - \hat{c}_2 - v)du\, dv. \tag{31}$$

Here $p_u, p_v$ represents the PDF of $u$ and $v$. The same is true for the integration regions $\Omega_3 = \{(b_1, b_2) : b_1 - b_2 < \hat{c}_1 - \hat{c}_2, \hat{c}_2 - \lambda < b_2 < \hat{c}_2\}$ and $\Omega_4 = \{(b_1, b_2) : b_1 - b_2 > \hat{c}_1 - \hat{c}_2, \hat{c}_2 - \lambda < b_2 < \hat{c}_2\}$, and the second term of integral can be expressed by

$$\iint_{\Omega_3} p_u(u)p_v(v) - p_u(u)p_v(\hat{c}_1 - \hat{c}_2 - v)du\, dv. \tag{32}$$

By Assumption (3) we have $p_u(u)p_v(v) > p_u(u)p_v(\hat{c}_1 - \hat{c}_2 - v)$, thus $p(\hat{c}_1 - \hat{c}_2 > b_1 - b_2) > \frac{1}{2}(p(\hat{c}_1 - \lambda > b_1) + p(\hat{c}_2 - \lambda < b_2))$. With Eq.(26), we finally get:

$$\mathbb{E}_{\mathcal{D}_{\mathrm{pref}}}\left[\mathbb{I}\left(\hat{H} \cdot [c(p, s_1^i) - c(p, s_2^i)] \geq 0\right)\right] > \mathbb{E}_{\mathcal{D}_{\mathrm{hard}}}\left[\mathbb{I}\left(\hat{H} \cdot [c(p, s^i) - \lambda] \geq 0\right)\right]. \tag{33}$$

$\square$

# B ADDITIONAL EXPERIMENT

## B.1 EXPERIMENT DETAILS

We employ the verl framework in (Sheng et al., 2024), where we use PyTorch FSDP to execute diverse RLHF dataflows and attain high throughput. The entire process of RL training includes

Table 5: The performance of policy model initialized by Qwen2.5-Math-1.5B trained with PRMs on GRPO.

|  | GSM8K | AMC | MATH | Olympiad Bench |
|---|---|---|---|---|
| Math-Shepherd-PRM-7B | 88.4 | 23.6 | 50.2 | 25.1 |
| EurusPRM-Stage2 | 87.7 | 22.2 | 49.6 | 23.8 |
| Skywork-PRM-7B | 88.2 | 23.8 | 50.2 | 25.3 |
| Math-PSA | 88.0 | 21.7 | 50.6 | 24.3 |
| PPRM | **88.6** | **24.7** | **51.0** | **25.7** |

Table 6: The performance of policy model initialized by Qwen2.5-Math-1.5B trained with PRMs on RLOO and GRPO with various advantage estimators.

|  | GSM8K | AMC | MATH | Olympiad Bench |
|---|---|---|---|---|
| RLOO | 87.8 | 25.8 | 49.6 | 24.5 |
| ReMax | 87.5 | 25.2 | 50.4 | 24.9 |
| GRPO | 88.6 | 24.7 | 51.0 | 25.7 |
| GRPO-P | **88.8** | **26.0** | **53.2** | **26.2** |

generating multiple candidate solutions to the instruction using the generation model, and scoring each candidate using a PRM. We use vLLM (Kwon et al., 2023) to implement the process. Our PPRM is trained on 4 A6000 GPUs and we performan RL training on 8 A6000 GPUs. We conduct RL training based on Qwen2.5-Math-1.5B and we report the result in Table 5 and 6. The training data consists of chain-of-thought format questions from the MATH dataset. For reward modeling, we compare our PPRM with Math-Shepherd, EurusPRM-Stage2, and MATH-PSA. For GRPO and RLOO implementation, we set the policy model learning rate to 1e-6 with a KL coefficient of 0.001. We generate 8 outputs per question with a maximum sequence length of 1024 tokens. The training batch size is configured as 128 to balance memory constraints and training efficiency.

### B.2 PROMPT TEMPLATE FOR LLM-AS-A-JUDGE

To construct PRM training data via LLM-as-a-judge, we use the following prompt, using the template in (Zhang et al., 2025).

---
**Prompt for constructing PRM training data via LLM-as-a-judge**

```
I will provide a math problem along with a solution.
They will be formatted as follows:

[Math Problem]

<math_problem>
...(math problem)...
</math_problem>

[Solution]

<paragraph_1>
...(paragraph 1 of solution)...
</paragraph_1>

...

<paragraph_n>
...(paragraph n of solution)...
</paragraph_n>
```
---

```
Your task is to review each paragraph of the solution in
sequence, analyzing, verifying, and critiquing the
reasoning in detail. You need to provide the
analyses and the conclusion in the following format:

<analysis_1>
...(analysis of paragraph 1)...
</analysis_1>

...

<analysis_n>
...(analysis of paragraph n)...
</analysis_n>

<conclusion>
Correct/Incorrect
</conclusion>

* When you analyze each paragraph, you should use proper
verification, recalculation, or reflection to indicate
whether it is logically and mathematically valid.
Please elaborate on the analysis process carefully.

* If an error is detected in any paragraph, you should describe
the nature and cause of the error in detail, and suggest
how to correct the error or the correct approach.
Once a paragraph is found to contain any error, stop
further analysis of subsequent paragraphs (as they may
depend on the identified error) and directly provide
the conclusion of "Incorrect."

For instance, given a solution of five paragraphs, if an error
is found in the third paragraph, you should reply in the
following format:

<analysis_1>
...(analysis of paragraph 1)...
</analysis_1>

<analysis_2>
...(analysis of paragraph 2)...
</analysis_3>

<analysis_3>
...(analysis of paragraph 3; since an error is found here, also
provide detailed critique and correction guideline)...
</analysis_3>

<conclusion>
Incorrect
</conclusion>

Note that the analyses of paragraphs 4 and 5 should be skipped
as the paragraph 3 has been found to contain an error.
```

Table 7: The performance of policy model initialized by Qwen2.5-Math-7B trained with PPRM and hard-label PRM. The experiments independently examine the impact of the new advantage estimator (with or without reference advantage estimator) and preference-based rewards on the performance of the policy model.

| Estimator | Reward Model | GSM8K | AMC | MATH | Olympiad Bench | AIME |
|-----------|--------------|-------|-----|------|----------------|------|
| w/ pref | PPRM | 95.7 | **48.6** | **76.6** | **55.6** | **19.1** |
| | Hard Label (MCTS) | 95.2 | 45.8 | 74.9 | 51.2 | 15.3 |
| w/o pref | PPRM | **95.9** | 47.7 | 75.6 | 53.4 | 16.2 |
| | Hard Label (MCTS) | 95.0 | 44.5 | 73.4 | 51.4 | 15.1 |

```
* Respond with your analyses and conclusion directly.

--------------------------------------------------

The following is the math problem and the solution for you task:

[Math Problem]

{tagged_problem}

[Solution]

{tagged_response}
```

## B.3 ABLATION STUDIES

We supplement additional ablation studies to isolate these factors. Specifically, we set the baseline as a PRM trained with hard labels annotated by MCTS (as used in Q1), and then separately introduce (i) PPRM using LLM-as-a-judge and (ii) the new advantage estimator into the RL training pipeline. We conduct RL training based on Qwen2.5-Math-7B. We set the policy model learning rate to 1e-6 with a KL coefficient of 0.001. During exploration, we generate 8 outputs per question with a maximum sequence length of 1024 tokens. The training batch size is configured as 128 to balance memory constraints and training efficiency.

We can find from the results in Table 7. The experiments independently examine the impact of the new advantage estimator (with or without reference advantage estimator) and preference-based rewards on the performance of the policy model. The results indicate that (i) Both the new advantage estimator and the use of preference-based rewards contribute to performance gains over the baseline RL setup. (ii) When applied on top of a hard-label PRM, the new advantage estimator still yields a modest improvement. This is because the hard-label PRM's output—interpreted as the estimated probability of a rollout being correct—can still benefit from a more refined advantage computation that accounts for a reference trajectory.

The term $\alpha$ ensures that higher-quality rollouts are more likely to be chosen, while the term $\beta$ penalizes overly complex solutions, favoring concise reasoning paths. In our supplementary experimental results, we find that the model is not sensitive to the parameters. We adjust $\alpha$ across [0.4,0.6] and $\beta$ across [0.8,0.95], and the preformance of the PPRM remains stable. However, when the parameters exceed this range, the performance of the reward model will deteriorate.

## B.4 RESULTS ON OTHER TASKS

We evaluate the resulting policies which apply PPRM (LLM-as-a-Judge) as the reward model in reinforcement learning on three diverse benchmarks: LeetCode (algorithmic coding), Live-CodeBench (real-world code generation), and MMLU-STEM (scientific and technical knowledge across physics, chemistry, biology, and engineering). The results in Table 8 show that policies trained with PPRM achieve state-of-the-art performance across these scientific and program synthe-

Table 8: The performance of policy model initialized by Qwen2.5-Math-7B trained with PRMs on GRPO.

| | AMC | MATH | OlympiadBench | AIME | LeetCode | LiveCodeBench | MMLU STEM |
|---|---|---|---|---|---|---|---|
| ORM | 38.84 | 70.78 | 49.87 | 10.31 | 23.42 | 22.12 | 80.23 |
| Math-Shepherd-PRM-7B | 44.47 | 74.03 | 52.46 | 16.71 | 25.76 | 24.33 | 84.63 |
| Math-PSA | 21.49 | 73.88 | 52.55 | 13.33 | 24.51 | 23.21 | 82.19 |
| Skywork-PRM-7B | 45.73 | 74.47 | 53.04 | 15.82 | 26.75 | 24.12 | 84.87 |
| EurusPRM-Stage2 | 44.49 | 73.80 | 51.15 | 16.24 | 25.75 | 24.21 | 84.45 |
| PPRM (Ours) | **47.97** | **76.44** | **56.01** | **18.87** | **27.22** | **26.03** | **85.14** |

Table 9: The performance of policy model initialized by Qwen2.5-Math-7B trained with PRMs on GRPO.

| | GSM8K | MATH | OlympiadBench | MMLU STEM |
|---|---|---|---|---|
| pass@8 | 95.8 | 88.7 | 57.2 | 80.3 |
| Hard-label PRM | 94.2 | 80.4 | 46.5 | 60.2 |
| PPRM | 94.4 | 80.2 | 47.7 | 61.5 |

sis tasks, consistently outperforming those trained with hard-label PRMs and other baseline reward models. This demonstrates that the benefits of preference-based process reward modeling—namely, reduced bias in automatic labeling and better alignment between intermediate steps and final outcomes—are not limited to mathematical reasoning, but generalize effectively to symbolic manipulation, scientific question answering, and code generation.

We further evaluate the PPRM with greedy search by generating $N$ candidate next steps at each reasoning step, scoring these candidates using the PRM, and selecting the highest-scoring step for subsequent expansion. For the policy model, we use Qwen2.5-7B-Instruct, which exhibits greater generative diversity, to sample 8 candidates at each step with sampling parameters set to temperature = 1.0 and top-p= 1.0. We conduct comparative experiments against the hard-label PRM using MCTS and LLM-as-a-Judge.

As shown in Table 9, PPRM combined with greedy search@8 achieves slightly better performance than the hard-label PRM. At each reasoning step, we select the candidate with the highest PRM-predicted score, where the score reflects the model's estimate of that step's correctness. However, this greedy selection based on local optimality does not guarantee a correct final answer, as the best-looking step at an intermediate stage may still lead the reasoning astray in subsequent steps. So we believe there remains substantial room for future exploration in designing more suitable search strategies—such as integrating step-level rewards with value estimates—to jointly account for both the correctness of the current reasoning step and the likelihood of ultimately arriving at a correct solution.

### B.5 PPRM USING DIFFERENT BACKBONES

we have extended our experiments to include two additional 7B-scale base models: Mistral-MetaMath-7B and DeepSeek-Math-7B-Base, training both hard-label PRMs and our PPRM using LLM-as-a-Judge on each. The results show that PPRM consistently outperforms the hard-label baseline across all three backbones, confirming the robustness of our preference-based approach.

As shown in Table 10, the relative gain from PPRM over hard-label PRM remains consistent, underscoring that the benefit of preference learning is not tied to a specific model family. When using DeepSeek-Math-7B-Base as the policy model, PPRM achieves performance comparable to that observed with Qwen2.5-Math-7B and both reward modeling variants yield lower absolute performance on Mistral-MetaMath-7B. These findings support our claim that PPRM is a general and backbone-agnostic reward modeling framework.

Table 10: Performance comparison of 7B reward models in PROCESSBENCH across GSM8K, MATH, OlympiadBench and Omni-MATH. The models are trained on the different data construction methods including MCTSa and LLM-as-a-judge and labeling methods including PPRM and hard label, and differen backbones.

| Method | Backbone | GSM8K | MATH | OlympiadBench | Omni-MATH | Avg. |
|---|---|---|---|---|---|---|
| PPRM | Qwen2.5-Math-7B-Instruct | 0.772 | 0.728 | 0.736 | 0.717 | 0.738 |
| | Deepseek-Math-7B-base | 0.751 | 0.735 | 0.732 | 0.715 | 0.733 |
| | Mistral-MetaMath-7b | 0.747 | 0.685 | 0.636 | 0.612 | 0.670 |
| Hard Label | Qwen2.5-Math-7B-Instruct | 0.770 | 0.709 | 0.682 | 0.678 | 0.710 |
| | Deepseek-Math-7B-base | 0.768 | 0.704 | 0.690 | 0.675 | 0.709 |
| | Mistral-MetaMath-7b | 0.744 | 0.661 | 0.616 | 0.603 | 0.655 |

