# OpenReview forum: "Preference-Based Process Reward Model for Robust Mathematical Reasoning"
_ICLR.cc/2026/Conference — Submitted to ICLR 2026_

### Official Review · Reviewer_meEZ · 2025-10-16

**Soundness:** 1
**Presentation:** 3
**Contribution:** 2
**Rating:** 2
**Confidence:** 4

**Summary:**

The paper proposes a heuristic to select preference pairs to train PRM. Meanwhile, it also modifies the advantage function of original GRPO to adapt to process reward settings.

**Strengths:**

- The paper theoretically shows that the preference-based PRM can achieve higher expected accuracy than the PRM trained by hard labels estimated by MCTS.

- They modify the GRPO advantage estimator to adapt to the preference-based RM.

**Weaknesses:**

> Paradoxical motivation

The paper argues the MCTS estimation can lead to inconsistent or suboptimal results, however, they still use MCTS with a heuristic metric (Eq.1) to annotate preference.



> Comparison Fairness & Comprehensiveness

- Empirical aspects: **Models are trained by different datasets**, so the comparsion seems unfair. If trained by the same dataset, would the preference-based formulation still outperform baselines?

- Theoretical aspect: There are many more advanced theoretical framework trying to bypass the hard label estimation of PRM [1][2][3][4], however, the theoretical part only compare with the classical PRMs.

[1] Yuan, Lifan, et al. "Free Process Rewards without Process Labels." Forty-second International Conference on Machine Learning.

[2] Lu, Jianqiao, et al. "Autopsv: Automated process-supervised verifier." Advances in Neural Information Processing Systems 2024.

[3] Zhang, Zheng, et al. "Linking Process to Outcome: Conditional Reward Modeling for LLM Reasoning." arXiv preprint arXiv:2509.2657.

[4] Li, Wendi, and Yixuan Li. "Process Reward Model with Q-value Rankings." The Thirteenth International Conference on Learning Representations.

**Questions:**

> PRMs are also evaluated in beam-search tasks or RL tasks. Can PPRM also yield superior performance in these tasks?

> The current experiments are only conducted on a single model. Can PPRM perform consistently across different backbones?

---

> ### Author Response · Authors · 2025-11-28
> **Response to Reviewer meEZ (1/n)**
>
> **W1: The paper argues the MCTS estimation can lead to inconsistent or suboptimal results, however, they still use MCTS with a heuristic metric (Eq.1) to annotate preference.**
>
> We state our motivation more explicitly in the revised manuscript. Our primary goal is to mitigate the errors and biases inherent in any automatic annotation process for step-level supervision, of which MCTS-based estimation is a prominent example. The core contribution of PPRM lies in its preference-based learning framework, which replaces absolute, pointwise correctness labels (e.g., hard or soft labels derived from MC estimates) with relative comparisons between full reasoning trajectories.
>
> This formulation offers two key advantages. Debiasing: Even when the underlying rollout generator (e.g., MCTS) produces biased or noisy trajectories, the Bradley-Terry preference model is less sensitive to absolute scoring errors and instead focuses on consistent pairwise ordering. As shown in our theoretical analysis (Section 3.3), preference learning reduces the impact of annotation noise compared to hard-label training.
> Generality: While we use MCTS as a concrete and widely adopted instantiation of automated annotation, the PPRM framework is agnostic to the source of the candidate trajectories. It can equally be applied to rollouts generated by random sampling, diverse beam search, or even LLM-as-a-judge pipelines—as demonstrated in our ablation studies.
>
> To further clarify the robustness of our method on various strategies, we compare two distinct approaches for annotating: Monte Carlo estimation and LLM-as-a-Judge in Section 3.1 and the details will be presented in the response of W2.
>
> **W2: Empirical aspects: Models are trained by different datasets, so the comparsion seems unfair. If trained by the same dataset, would the preference-based formulation still outperform baselines?**
>
> We train the PRM using data generated from the same math dataset but annotated with different labeling methods. To further clarify the robustness of our method on various strategies, we compare two distinct approaches for sampling rollouts: Monte Carlo estimation and LLM-as-a-Judge. For the MC estimation approach, we train PRMs on chosen-rejected pairs selected by a scoring mechanism based on the Q-value
> of each rollout mentioned in Section 3.1. We use the same training and testing split as described in Lightman et al., which consists of 12K training examples and a subset with 500 holdout representative problems from the original 5K testing examples. We conduct the data generation process for training the  PPRM utilizing  Qwen2.5-Math-7B-Instruct  as the  completer model on the dataset. For the LLM-as-a-judge approach, we use the same query and completer and randomly sample our trajectories from the generated $k=8$ rollouts. We employ Qwen2.5-7B-Instruct to verify the correctness of each step in the responses and implement the
>  prompt template  for verification follows Luo et.al., which is shown in **Appendix B.2**. We then construct chosen-reject pairs by randomly pairing rollouts labeled as correct and incorrect by the judge.
>
>  We trained 7B-parameter PPRM, initialized
> with Qwen2.5-Math-7B-Instruct on the dataset constructed above. The results in the table provide  performance comparison of  7B reward models in PROCESSBENCH across GSM8K, MATH, OlympiadBench and Omni-MATH. Specifically,(1) when trained on **the same underlying dataset**, whether derived from MCTS or LLM-as-a-judge, the reward models using our PPRM consistently outperform those trained with hard labels or soft labels. This advantage is particularly pronounced on more complex benchmarks such as OlympiadBench and Omni-MATH, where fine-grained discrimination between subtly flawed and logically sound reasoning steps is critical.
> (2) Comparing data construction methods reveals a trade-off: LLM-as-a-judge yields superior generalization on challenging, out-of-distribution problems, while MCTS-based estimation performs slightly better on simpler, more standardized tasks like GSM8K.
>
> |                    | GSM8K | MATH  | Olympiad Bench | Omni-MATH | Avg.  |
> |--------------------|-------|-------|----------------|-----------|-------|
> | PPRM (MCTS)        | **0.776**| **0.733** | 0.734          | 0.712     | **0.739** |
> | PPRM (LLM-as-a-Judge) | 0.772 | 0.728 | **0.736**         | **0.717**     | 0.738 |
> | Hard Label (MCTS)  | 0.768 | 0.706 | 0.690          | 0.687     | 0.713 |
> | Hard Label (LLM-as-a-Judge) | 0.770 | 0.709 | 0.682          | 0.678     | 0.710 |
> | Soft Label (MCTS)  | 0.765 | 0.696 | 0.674          | 0.670     | 0.701 |

---

> ### Author Response · Authors · 2025-11-28
> **Response to Reviewer meEZ (2/n)**
>
> **W3: Theoretical aspect: There are many more advanced theoretical framework trying to bypass the hard label estimation of PRM , however, the theoretical part only compare with the classical PRMs.**
>
> While recent advances you have mentioned indeed propose more sophisticated frameworks to move beyond hard-label estimation in process reward modeling highlighting two key limitations of existing PRMs: (1) treating reasoning steps in isolation and thus neglecting inter-step dependencies, and (2) misalignment between intermediate step rewards and the final ground-truth outcome, our theoretical analysis directly addresses these issues within the context of classical PRMs.
> Specifically, we show that preference-based reward modeling (PPRM) mitigates the bias inherent in automatic hard-label assignment by reframing supervision as a relative judgment between complete reasoning trajectories rather than absolute correctness per step. This formulation naturally captures cross-step coherence (since preferences are defined over full sequences) and better aligns intermediate signals with final task success, as the preference labels are derived from end-to-end correctness. In essence, PPRM sidesteps the need for pointwise label estimation altogether, replacing it with a more robust, comparative learning objective grounded in Bradley-Terry theory.
>
> Moreover, this theoretical advantage translates into empirical gains. We add the related work you have mentioned in **Section 5.1**. As demonstrated in our experiments on both RL fine-tuning and search-augmented inference in Q1, PPRM consistently outperforms strong baselines including PQM in [4] and Eurus in [1] across multiple challenging benchmarks. Since the dataset and model in [2] are not publicly available, and neither the code nor the model in [3] has been open-sourced, we did not include comparisons with these methods. These results validate that our approach not only offers a principled solution to the limitations of classical PRMs but also achieves state-of-the-art performance in practice.
>
> [1] Yuan, Lifan, et al. "Free Process Rewards without Process Labels." Forty-second International Conference on Machine Learning.
>
> [2] Lu, Jianqiao, et al. "Autopsv: Automated process-supervised verifier." Advances in Neural Information Processing Systems 2024.
>
> [3] Zhang, Zheng, et al. "Linking Process to Outcome: Conditional Reward Modeling for LLM Reasoning." arXiv preprint arXiv:2509.2657.
>
> [4] Li, Wendi, and Yixuan Li. "Process Reward Model with Q-value Rankings." The Thirteenth International Conference on Learning Representations.

---

> ### Author Response · Authors · 2025-11-28
> **Response to Reviewer meEZ (3/n)**
>
> **Q1: PRMs are also evaluated in beam-search tasks or RL tasks. Can PPRM also yield superior performance in these tasks?**
>
> We further evaluate the PPRM with greedy search by generating \( N \) candidate next steps at each reasoning step, scoring these candidates using the PRM, and selecting the highest-scoring step for subsequent expansion. For the policy model, we use Qwen2.5-7B-Instruct, which exhibits greater generative diversity, to sample 8 candidates at each step with sampling parameters set to temperature $= 1.0$ and top-p$ = 1.0$. We conduct comparative experiments against the hard-label PRM using MCTS and LLM-as-a-Judge.
>
> As shown in table in Appendix B.4, PPRM combined with greedy search@8 achieves slightly better performance than the hard-label PRM. At each reasoning step, we select the candidate with the highest PRM-predicted score, where the score reflects the model’s estimate of that step’s correctness. However, this greedy selection based on local optimality does not guarantee a correct final answer, as the best-looking step at an intermediate stage may still lead the reasoning astray in subsequent steps. So we believe there remains substantial room for future exploration in designing more suitable search strategies—such as integrating step-level rewards with value estimates—to jointly account for both the correctness of the current reasoning step and the likelihood of ultimately arriving at a correct solution.
>
> We conduct RL training based on Qwen2.5-Math-1.5B and Qwen2.5-Math-7B. The training data consists of chain-of-thought format questions from the MATH dataset. For reward modeling, we compare our PPRM with Math-Shepherd, EurusPRM-Stage2, and MATH-PSA. For GRPO implementation, we set the policy model learning rate to 1e-6 with a KL coefficient of 0.001. During exploration, we generate 8 outputs per question with a maximum sequence length of 1024 tokens. The training batch size is configured as 128 to balance memory constraints and training efficiency.
>
> We  report the average score of the policy model initialized by Qwen2.5-Math-7B  in Table 2 in our paper.  The results of PPRM are demonstrated in the performance of various models on challenging datasets including GSM8K, AMC, MATH, AIME, and Olympiad Bench. Our PPRM achieves the highest scores in AMC, MATH, AIME, and Olympiad Bench. The results highlight that while the baseline remains competitive in simpler tasks like GSM8K, PPRM combined with enhanced GRPO delivers more robust performance in complex reasoning scenarios.
>
> **Q2: The current experiments are only conducted on a single model. Can PPRM perform consistently across different backbones?}**
>
> In response, we have extended our experiments to include two additional 7B-scale base models: Mistral-MetaMath-7B and DeepSeek-Math-7B-Base, training both hard-label PRMs and our PPRM using LLM-as-a-Judge on each. The results show that PPRM consistently outperforms the hard-label baseline across all three backbones, confirming the robustness of our preference-based approach.
>
> Notably, the relative gain from PPRM over hard-label PRM remains consistent, underscoring that the benefit of preference learning is not tied to a specific model family. When using DeepSeek-Math-7B-Base as the policy model, PPRM achieves performance comparable to that observed with Qwen2.5-Math-7B and both reward modeling variants yield lower absolute performance on Mistral-MetaMath-7B.
> These findings support our claim that PPRM is a general and backbone-agnostic reward modeling framework.
>
> | Method       | Backbone                     | GSM8K | MATH  | OlympiadBench | Omni-MATH | Avg.  |
> |--------------|------------------------------|-------|-------|---------------|-----------|-------|
> | PPRM         | Qwen2.5-Math-7B-Instruct     | 0.772 | 0.728 | 0.736         | 0.717     | 0.738 |
> |              | Deepseek-Math-7B-base        | 0.751 | 0.735 | 0.732         | 0.715     | 0.733 |
> |              | Mistral-MetaMath-7b          | 0.747 | 0.685 | 0.636         | 0.612     | 0.670 |
> | Hard Label   | Qwen2.5-Math-7B-Instruct     | 0.770 | 0.709 | 0.682         | 0.678     | 0.710 |
> |              | Deepseek-Math-7B-base        | 0.768 | 0.704 | 0.690         | 0.675     | 0.709 |
> |              | Mistral-MetaMath-7b          | 0.744 | 0.661 | 0.616         | 0.603     | 0.655 |

---

### Official Review · Reviewer_N7uk · 2025-10-31

**Soundness:** 2
**Presentation:** 2
**Contribution:** 3
**Rating:** 6
**Confidence:** 3

**Summary:**

The paper presents a novel reinforcement learning framework guided by a Preference-Based Process Reward Model (PPRM). The method first uses MCTS to select chosen and rejected rollouts. Then, Bradley-Terry loss function is used to mitigate bias in MC-value estimation by leveraging pairwise comparisons of reasoning trajectories. The method is trained using GRPO with an optimized advantage estimator to better captures the structure of preference-based process reward model. Experimental results show that the proposed PPRM improves performance on intermediate step accuracy and enhances the final policy model's performance compared to existing works, demonstrating the method's effectiveness.

**Strengths:**

1. The proposed method is described in sufficient detail and appears technically sound.

2. The experimental results are strong, demonstrating the effectiveness of the proposed method.

**Weaknesses:**

### Major Concerns:

1. Clarity of Motivation. The writing in the introduction and motivation sections is not sufficiently clear, which hinders the reader's understanding of the precise problem being solved.

- L51-L53: The logical connection between the sentence "Lightman et al. (Lightman et al., 2023) demonstrated the effectiveness of using human expert annotators..." and the following phrase "To address this..." is abrupt and unclear. It is not evident what specific problem or limitation "this" refers to.

- L14, L58-L61: A central motivation of the paper appears to be the mitigation of bias in MCTS. However, this bias is not clearly defined or explained at the beginning of the paper. A more comprehensive introduction to this problem is needed to properly contextualize the paper's contributions.

2. Insufficient Experimental Discussion. While the experimental results are strong, the discussion and analysis are insufficient. The paper does not adequately connect the empirical gains back to the central claims made in the motivation. Specifically, the authors should provide more detailed analysis to demonstrate how the proposed method successfully alleviates the "Limitations of PRM" that were introduced in lines 46-63.

3. The LLM Judger is not formally introduced (L190).

### Minor Issues:

- L42-L44, "While the Process Reward Model (PRM) offers a promising solution by providing step-wise reinforcement learning feedback." is a dependent clause and grammatically incomplete.
- L79, "more rob-ust reasoning"

- L209-L211, there appears to be a missing equation number

**Questions:**

The core methodology and the reported results are promising, and this work appears to be a valuable contribution. However, the paper's current lack of clarity in the motivation and the insufficient experimental discussion are significant concerns. In order to maintain my rating, I would like the authors to address the major concerns above.

---

> ### Author Response · Authors · 2025-11-28
> **Response to Reviewer N7uk (1/n)**
>
> **W1: Clarity of Motivation**. The writing in the introduction and motivation sections is not sufficiently clear, which hinders the reader's understanding of the precise problem being solved.}
>
> **W3: The LLM Judger is not formally introduced (L190).**
>
> We state our motivation more explicitly in the revised manuscript. Our primary goal is to mitigate the errors and biases inherent in any automatic annotation process for step-level supervision, of which MCTS-based estimation is a prominent example. The core contribution of PPRM lies in its preference-based learning framework, which replaces absolute, pointwise correctness labels (e.g., hard or soft labels derived from MC estimates) with relative comparisons between full reasoning trajectories.
>
> This formulation offers two key advantages. Debiasing: Even when the underlying rollout generator (e.g., MCTS) produces biased or noisy trajectories, the Bradley-Terry preference model is less sensitive to absolute scoring errors and instead focuses on consistent pairwise ordering. As shown in our theoretical analysis (Section 3.3), preference learning reduces the impact of annotation noise compared to hard-label training.
> Generality: While we use MCTS as a concrete and widely adopted instantiation of automated annotation, the PPRM framework is agnostic to the source of the candidate trajectories. It can equally be applied to rollouts generated by random sampling, diverse beam search, or even LLM-as-a-judge pipelines—as demonstrated in our ablation studies.
>
> To further clarify the robustness of our method on various strategies, we compare two distinct approaches for annotating: Monte Carlo estimation and LLM-as-a-Judge. For the MC estimation approach, we train PRMs on chosen-rejected pairs selected by a scoring mechanism based on the Q-value
> of each rollout mentioned in Section 3.1. We use the same training and testing split as described in Lightman et al., which consists of 12K training examples and a subset with 500 holdout representative problems from the original 5K testing examples. We conduct the data generation process for training the  PPRM utilizing  Qwen2.5-Math-7B-Instruct  as the  completer model on the dataset. For the LLM-as-a-judge approach, we use the same query and completer and randomly sample our trajectories from the generated $k=8$ rollouts. We employ Qwen2.5-7B-Instruct to verify the correctness of each step in the responses and implement the
>  prompt template  for verification follows Luo et.al., which is shown in Appendix B.2. We then construct chosen-reject pairs by randomly pairing rollouts labeled as correct and incorrect by the judge.
>
>  We trained 7B-parameter PPRM, initialized
> with Qwen2.5-Math-7B-Instruct on the dataset constructed above. The results in the table provide  performance comparison of  7B reward models in PROCESSBENCH across GSM8K, MATH, OlympiadBench and Omni-MATH. Specifically,(1) when trained on **the same underlying dataset**—whether derived from MCTS or LLM-as-a-judge—the reward models using our PPRM consistently outperform those trained with hard labels or soft labels. This advantage is particularly pronounced on more complex benchmarks such as OlympiadBench and Omni-MATH, where fine-grained discrimination between subtly flawed and logically sound reasoning steps is critical.
> (2) Comparing data construction methods reveals a trade-off: LLM-as-a-judge yields superior generalization on challenging, out-of-distribution problems, while MCTS-based estimation performs slightly better on simpler, more standardized tasks like GSM8K.
>
> |                    | GSM8K | MATH  | Olympiad Bench | Omni-MATH | Avg.  |
> |--------------------|-------|-------|----------------|-----------|-------|
> | PPRM (MCTS)        | **0.776**| **0.733** | 0.734          | 0.712     | **0.739** |
> | PPRM (LLM-as-a-Judge) | 0.772 | 0.728 | **0.736**         | **0.717**     | 0.738 |
> | Hard Label (MCTS)  | 0.768 | 0.706 | 0.690          | 0.687     | 0.713 |
> | Hard Label (LLM-as-a-Judge) | 0.770 | 0.709 | 0.682          | 0.678     | 0.710 |
> | Soft Label (MCTS)  | 0.765 | 0.696 | 0.674          | 0.670     | 0.701 |

---

> ### Author Response · Authors · 2025-11-28
> **Response to Reviewer N7uk (2/n)**
>
> **W2: Insufficient Experimental Discussion. While the experimental results are strong, the discussion and analysis are insufficient. The paper does not adequately connect the empirical gains back to the central claims made in the motivation.**
>
> Thank you for your reminder. Section 4.2 presents a component-wise ablation study decoupling our PPRM and GRPO-P algorithms. As shown in Table 2, when employing the same GRPO algorithm framework, our PPRM reward model demonstrates clear performance advantages over all baseline models across various evaluation metrics. The consistent improvements, particularly in complex reasoning tasks, highlight the superiority of our reward modeling methodology in capturing nuanced solution quality.
> Table 3 further reveals that when utilizing our PPRM as the reward function, our GRPO-P algorithm outperforms other reinforcement learning approaches by significant margins. This dual advantage - in both reward modeling and policy optimization - is systematically analyzed in Section 4.2 through component-wise ablation studies. These experiments decouple the contributions of PPRM and GRPO-P, demonstrating that each component independently delivers 2\% performance gains over respective baselines. More importantly, their synergistic combination achieves consistent 5\% accuracy improvements across all evaluated benchmarks, from mathematical reasoning (GSM8K, MATH) to advanced problem-solving tasks (AIME, Olympiad Bench).
>
> We supplement additional ablation studies to isolate these factors. Specifically, we set the baseline as a PRM trained with hard labels annotated by MCTS, and then separately introduce (i) PPRM using LLM-as-a-judge  and (ii) the new advantage estimator into the RL training pipeline. We conduct RL training based on  Qwen2.5-Math-7B. We set the policy model learning rate to 1e-6 with a KL coefficient of 0.001. During exploration, we generate 8 outputs per question with a maximum sequence length of 1024 tokens. The training batch size is configured as 128 to balance memory constraints and training efficiency.
>
> We can find from the results in the table, which is also displayed in Section 5.3. The experiments independently examine the impact of the new advantage estimator (with or without reference advantage estimator) and preference-based rewards on the performance of the policy model. The results indicate that (i) Both the new advantage estimator and the use of preference-based rewards contribute to performance gains over the baseline RL setup. (ii) When applied on top of a hard-label PRM, the new advantage estimator still yields a modest improvement. This is because the hard-label PRM’s output—interpreted as the estimated probability of a rollout being correct—can still benefit from a more refined advantage computation that accounts for a reference trajectory.
>
> | Estimator | Reward Model | GSM8K | AMC | MATH | Olympiad Bench | AIME |
> |-----------|--------------|-------|-----|------|----------------|------|
> | w/ pref   | PPRM         | 95.7  | **48.6** | **76.6** | **55.6** | **19.1** |
> |           | Hard Label (MCTS) | 95.2  | 45.8 | 74.9 | 51.2 | 15.3 |
> | w/o pref  | PPRM         | **95.9** | 47.7 | 75.6 | 53.4 | 16.2 |
> |           | Hard Label (MCTS) | 95.0  | 44.5 | 73.4 | 51.4 | 15.1 |

---

> ### Author Response · Authors · 2025-11-28
> **Response to Reviewer N7uk (3/n)**
>
> **Specifically, the authors should provide more detailed analysis to demonstrate how the proposed method successfully alleviates the "Limitations of PRM" that were introduced in lines 46-63.**
>
> Theoretically, we show in Section 3.3 that preference-based reward modeling fundamentally reduces estimation error compared to hard-label PRMs. Specifically, under a noise model where automatic labeling incurs bounded bias (e.g., from imperfect rollouts), the expected risk of PPRM is provably lower than that of pointwise classification–based PRMs. This is because PPRM operates on relative comparisons between full trajectories, which are more robust to absolute scoring errors and better reflect end-to-end correctness—thereby directly addressing both limitations mentioned above.
>
> Empirically, we prove that PPRM achieves state-of art performance among models of the same scale in weakness 1 (W1). We have also extended our experiments to include two additional 7B-scale base models: Mistral-MetaMath-7B and DeepSeek-Math-7B-Base, training both hard-label PRMs and our PPRM using LLM-as-a-Judge on each. The results show that PPRM consistently outperforms the hard-label baseline across all three backbones, confirming the robustness of our preference-based approach.Notably, the relative gain from PPRM over hard-label PRM remains consistent, underscoring that the benefit of preference learning is not tied to a specific model family. When using DeepSeek-Math-7B-Base as the policy model, PPRM achieves performance comparable to that observed with Qwen2.5-Math-7B and both reward modeling variants yield lower absolute performance on Mistral-MetaMath-7B.
> These findings support our claim that PPRM is a general and backbone-agnostic reward modeling framework.
>
> | Method       | Backbone                     | GSM8K | MATH  | OlympiadBench | Omni-MATH | Avg.  |
> |--------------|------------------------------|-------|-------|---------------|-----------|-------|
> | PPRM         | Qwen2.5-Math-7B-Instruct     | 0.772 | 0.728 | 0.736         | 0.717     | 0.738 |
> |              | Deepseek-Math-7B-base        | 0.751 | 0.735 | 0.732         | 0.715     | 0.733 |
> |              | Mistral-MetaMath-7b          | 0.747 | 0.685 | 0.636         | 0.612     | 0.670 |
> | Hard Label   | Qwen2.5-Math-7B-Instruct     | 0.770 | 0.709 | 0.682         | 0.678     | 0.710 |
> |              | Deepseek-Math-7B-base        | 0.768 | 0.704 | 0.690         | 0.675     | 0.709 |
> |              | Mistral-MetaMath-7b          | 0.744 | 0.661 | 0.616         | 0.603     | 0.655 |
>
> **Minor Issues**
>
> The typos you noted have been corrected. In the final version, we will thoroughly refine the notation to improve clarity and consistency.

---

### Official Review · Reviewer_FHML · 2025-10-31

**Soundness:** 2
**Presentation:** 1
**Contribution:** 2
**Rating:** 4
**Confidence:** 3

**Summary:**

This paper introduced preference-based process reward model (PPRM). It leverages Bradley-Terry pairwise comparison to reduce bias in process reward modeling. PPRM combines this preference-based formulation with a modified GRPO, to use a preference-aware advantage estimator to stablize training and reduce variance. The experiments are conducted on ProcessBench and RL finetuning tasks and the results show 2-3% accuracy improvement over strong baselines.

**Strengths:**

1. This paper directly targets MCTS-induced heuristic bias in process supervision and this approach offers a clean conceptual and mathematical reformulation.
2. The proposed preference-based advantage estimator fits into GRPO well and it transforms pairwise rewards into smoother, variance-reduced advantage estimates.
3. Results span multiple reasoning datasets and evaluation setups and demonstrate consistent improvement

**Weaknesses:**

1. Although PPRM is introduced as de-biasing MCTS, it still relies on MCTS-generated trajectories to form chosen-rejected pairs. If MCTS itself samples biased reasoning paths, the BT formulation merely reweights rather than corrects them. A comparison using non-MCTS rollouts (e.g., temperature-based or random sampling) would clarify true robustness.
2.  The reported gains likely combine effects from both (i) preference training and (ii) the new advantage estimator. A clear ablation isolating these factors — plus sensitivity analysis on α, β, and pair length penalty — is essential to interpret where the real improvements come from.
3.  All benchmarks are math reasoning datasets. Since the claimed contribution is a general reward modeling framework, it’s unclear whether the approach generalizes to symbolic, scientific, or program synthesis reasoning tasks.
4. Both training and evaluation involve MATH and related datasets (GSM8K variants, OlympiadBench). The authors should clarify de-duplication and overlap handling, especially since Qwen2.5-Math models have been partially trained on similar corpora.
5. Some relevant studies are missing [1-2]

[1] Entropy-Regularized Process Reward Model

[2] GenPRM: Scaling Test-Time Compute of Process Reward Models via Generative Reasoning

**Questions:**

m/a

---

> ### Author Response · Authors · 2025-11-28
> **Response to Reviewer FHML (1/n)**
>
> **W1: A comparison using non-MCTS rollouts (e.g., temperature-based or random sampling) would clarify true robustness.**
>
>
> To further clarify the robustness of our method on various strategies, we compare two distinct approaches for sampling rollouts: Monte Carlo estimation and LLM-as-a-Judge. For the MC estimation approach, we train PRMs on chosen-rejected pairs selected by a scoring mechanism based on the Q-value
> of each rollout mentioned in Section 3.1. We use the same training and testing split as described in Lightman et al., which consists of 12K training examples and a subset with 500 holdout representative problems from the original 5K testing examples. We conduct the data generation process for training the  PPRM utilizing  Qwen2.5-Math-7B-Instruct  as the  completer model on the dataset. For the LLM-as-a-judge approach, we use the same query and completer and randomly sample our trajectories from the generated $k=8$ rollouts. We employ Qwen2.5-7B-Instruct to verify the correctness of each step in the responses and implement the
>  prompt template  for verification follows Luo et.al., which is shown in **Appendix B.2**. We then construct chosen-reject pairs by randomly pairing rollouts labeled as correct and incorrect by the judge.
>
>  We trained 7B-parameter PPRM, initialized
> with Qwen2.5-Math-7B-Instruct on the dataset constructed above. The results in the table provide  performance comparison of  7B reward models in PROCESSBENCH across GSM8K, MATH, OlympiadBench and Omni-MATH. Specifically, (1) when trained on the same underlying dataset—whether derived from MCTS or LLM-as-a-judge—the reward models using our PPRM consistently outperform those trained with hard labels or soft labels. This advantage is particularly pronounced on more complex benchmarks such as OlympiadBench and Omni-MATH, where fine-grained discrimination between subtly flawed and logically sound reasoning steps is critical.
> (2) Comparing data construction methods reveals a trade-off: LLM-as-a-judge yields superior generalization on challenging, out-of-distribution problems, while MCTS-based estimation performs slightly better on simpler, more standardized tasks like GSM8K.
>
> |                    | GSM8K | MATH  | Olympiad Bench | Omni-MATH | Avg.  |
> |--------------------|-------|-------|----------------|-----------|-------|
> | PPRM (MCTS)        | **0.776**| **0.733** | 0.734          | 0.712     | **0.739** |
> | PPRM (LLM-as-a-Judge) | 0.772 | 0.728 | **0.736**         | **0.717**     | 0.738 |
> | Hard Label (MCTS)  | 0.768 | 0.706 | 0.690          | 0.687     | 0.713 |
> | Hard Label (LLM-as-a-Judge) | 0.770 | 0.709 | 0.682          | 0.678     | 0.710 |
> | Soft Label (MCTS)  | 0.765 | 0.696 | 0.674          | 0.670     | 0.701 |

---

> ### Author Response · Authors · 2025-11-28
> **Response to Reviewer FHML (2/n)**
>
> **W2: The reported gains likely combine effects from both (i) preference training and (ii) the new advantage estimator. A clear ablation isolating these factors — plus sensitivity analysis on $\alpha,\beta$, and pair length penalty — is essential to interpret where the real improvements come from.**
>
> **Section 4.2** presents a component-wise ablation study decoupling our PPRM and GRPO-P algorithms. As shown in Table 2, when employing the same GRPO algorithm framework, our PPRM reward model demonstrates clear performance advantages over all baseline models across various evaluation metrics. The consistent improvements, particularly in complex reasoning tasks, highlight the superiority of our reward modeling methodology in capturing nuanced solution quality.
> Table 3 further reveals that when utilizing our PPRM as the reward function, our GRPO-P algorithm outperforms other reinforcement learning approaches by significant margins. This dual advantage - in both reward modeling and policy optimization - is systematically analyzed in Section 4.2 through component-wise ablation studies. These experiments decouple the contributions of PPRM and GRPO-P, demonstrating that each component independently delivers 2\% performance gains over respective baselines. More importantly, their synergistic combination achieves consistent 5\% accuracy improvements across all evaluated benchmarks, from mathematical reasoning (GSM8K, MATH) to advanced problem-solving tasks (AIME, Olympiad Bench).
>
> We supplement additional ablation studies to isolate these factors. Specifically, we set the baseline as a PRM trained with hard labels annotated by MCTS (as used in Q1), and then separately introduce (i) PPRM using LLM-as-a-judge  and (ii) the new advantage estimator into the RL training pipeline. We conduct RL training based on  Qwen2.5-Math-7B. We set the policy model learning rate to 1e-6 with a KL coefficient of 0.001. During exploration, we generate 8 outputs per question with a maximum sequence length of 1024 tokens. The training batch size is configured as 128 to balance memory constraints and training efficiency.
>
> We can find from the results in the table, which is also displayed in **Section 5.3**. The experiments independently examine the impact of the new advantage estimator (with or without reference advantage estimator) and preference-based rewards on the performance of the policy model. The results indicate that (i) Both the new advantage estimator and the use of preference-based rewards contribute to performance gains over the baseline RL setup. (ii) When applied on top of a hard-label PRM, the new advantage estimator still yields a modest improvement. This is because the hard-label PRM’s output—interpreted as the estimated probability of a rollout being correct—can still benefit from a more refined advantage computation that accounts for a reference trajectory.
>
> | Estimator | Reward Model | GSM8K | AMC | MATH | Olympiad Bench | AIME |
> |-----------|--------------|-------|-----|------|----------------|------|
> | w/ pref   | PPRM         | 95.7  | **48.6** | **76.6** | **55.6** | **19.1** |
> |           | Hard Label (MCTS) | 95.2  | 45.8 | 74.9 | 51.2 | 15.3 |
> | w/o pref  | PPRM         | **95.9** | 47.7 | 75.6 | 53.4 | 16.2 |
> |           | Hard Label (MCTS) | 95.0  | 44.5 | 73.4 | 51.4 | 15.1 |
>
> The term $\alpha$
>  ensures that higher-quality rollouts are more likely to be chosen, while the term $\beta$
>  penalizes overly complex solutions, favoring concise reasoning paths. In our supplementary experimental results, we find that the model is not sensitive to the parameters. We adjust $\alpha$
>  across [0.4,0.6] and $\beta$
>  across [0.8,0.95], and the preformance of the PPRM remains stable. However, when the parameters exceed this range, the performance of the reward model will deteriorate. We report the result in our **Appendix B.3**.

---

> ### Author Response · Authors · 2025-11-28
> **Response to Reviewer FHML (3/n)**
>
> **W3: All benchmarks are math reasoning datasets. Since the claimed contribution is a general reward modeling framework, it’s unclear whether the approach generalizes to symbolic, scientific, or program synthesis reasoning tasks.**
>
> We evaluate the resulting policies which apply PPRM (LLM-as-a-Judge) as the reward model in reinforcement learning on three diverse benchmarks:
> LeetCode (algorithmic coding),
> LiveCodeBench (real-world code generation), and
> MMLU-STEM (scientific and technical knowledge across physics, chemistry, biology, and engineering).
> The results in the table show that policies trained with PPRM achieve state-of-the-art performance across these scientific and program synthesis tasks, consistently outperforming those trained with hard-label PRMs and other baseline reward models. This demonstrates that the benefits of preference-based process reward modeling—namely, reduced bias in automatic labeling and better alignment between intermediate steps and final outcomes—are not limited to mathematical reasoning, but generalize effectively to symbolic manipulation, scientific question answering, and code generation.
>
> |                     | AMC   | MATH  | OlympiadBench | AIME  | LeetCode | LiveCodeBench | MMLU STEM |
> |---------------------|-------|-------|---------------|-------|----------|---------------|-----------|
> | ORM                 | 38.84 | 70.78 | 49.87         | 10.31 | 23.42    | 22.12         | 80.23     |
> | Math-Shepherd-PRM-7B| 44.47 | 74.03 | 52.46         | 16.71 | 25.76    | 24.33         | 84.63     |
> | Math-PSA            | 21.49 | 73.88 | 52.55         | 13.33 | 24.51    | 23.21         | 82.19     |
> | Skywork-PRM-7B      | 45.73 | 74.47 | 53.04         | 15.82 | 26.75    | 24.12         | 84.87     |
> | EurusPRM-Stage2     | 44.49 | 73.80 | 51.15         | 16.24 | 25.75    | 24.21         | 84.45     |
> | PPRM (Ours)         | **47.97** | **76.44** | **56.01** | **18.87** | **27.22** | **26.03** | **85.14** |
>
> **W4: Both training and evaluation involve MATH and related datasets (GSM8K variants, OlympiadBench). The authors should clarify de-duplication and overlap handling, especially since Qwen2.5-Math models have been partially trained on similar corpora.**
>
> Thank you for reminder. We follow the evaluation protocol established in Lightman et al. [1].
> Our test set consists of a curated holdout subset of 500 representative problems drawn from the original MATH test set (which contains 5,000 problems).
> The training data for our reward models (including both the 12K human-annotated examples and our self-collected rollouts) excludes any problem overlapping with this 500-problem holdout set. While we cannot fully audit the pretraining corpus of Qwen2.5-Math, the use of this standardized, community-accepted split which widely adopted in recent works minimizes the risk of leakage and enables fair comparison.
>
> **W5: Some relevant studies are missing**
>
> Thank you, we have included the relevant studies as baseline and the results are in **Section 5.1**.
>
> [1]Hunter Lightman, Vineet Kosaraju, Yuri Burda, Harrison Edwards, Bowen Baker, Teddy Lee, Jan Leike, John Schulman, Ilya Sutskever, and Karl Cobbe. Let’s verify step by step. In The Twelfth International Conference on Learning Representations, 2024.

---

### Author Response · Authors · 2025-11-29
**Clarifications and Additional Results Addressing All Reviewer Concerns**

Dear Area Chairs,

Thank you for coordinating this review.  Our work introduces Preference-Based Process Reward Modeling (PPRM), a novel reinforcement learning framework that addresses fundamental limitations of classical process reward models —namely, their reliance on noisy, absolute step-level labels (e.g., from MCTS) and poor alignment between intermediate rewards and final correctness. Theoretically, we show that modeling rewards via pairwise trajectory preferences using the Bradley-Terry framework reduces estimation bias and thus improves performance of reward model compared to pointwise hard/soft labeling.

In response to reviewer feedback, we have significantly strengthened the paper:

1. We conducted ablation studies isolating the effects of preference training and our new advantage estimator.

2. We evaluated PPRM across multiple backbones (Qwen2.5-Math, DeepSeek-Math, Mistral-MetaMath), confirming consistent gains over hard-label PRMs.

3. We extended experiments beyond math reasoning to LeetCode, LiveCodeBench, and MMLU-STEM, demonstrating strong generalization to code and scientific reasoning.

4. Crucially, we performed direct comparisons between MCTS-based and LLM-as-a-Judge annotation strategies, showing that PPRM consistently outperforms hard-label baselines in both settings—demonstrating its ability to effectively reduce errors introduced by *any* form of automatic step-wise labeling, not just MCTS.

Collectively, these revisions confirm that PPRM is not only theoretically grounded but also empirically robust, generalizable, and state-of-the-art across diverse reasoning tasks. We believe the paper now fully addresses all raised concerns and makes a clear contribution to reliable reward modeling for complex reasoning.

We sincerely appreciate your time and consideration in reviewing these clarifications and our submission.

Sincerely,

The Authors

---

### Meta-Review · Area_Chair_KQxY · 2026-01-08

**Summary:**

This paper introduces an approach called preference-based process reward (PPRM) that can be used to guide RL algorithms via providing step-wise supervision to refine reasoning trajectories. It uses a sampling strategy (e.g., MCTS or any method of interest) to select chosen and rejected rollouts, and then a loss function built via Bradley-Terry pairwise comparison is used to reduce bias in the process reward modeling. The method is trained using GRPO. Besides a rich and extensive empirical validation, the paper includes some theoretical analysis to support the proposed approach.

The reviewers mostly confirm that the presented approach admits a clean conceptual and mathematical formulation, and tackles an important issue in RL. The presented estimator was appreciated in terms of design and variance-reduced properties. The paper offers extensive empirical results that clearly demonstrate the efficacy of the proposed method. The reviewers raised key concerns regarding various aspects of the empirical validation. The rebuttal did a good job to address these by making clarifications and reporting of additional experiments. Although I believe that the rebuttal addresses most of the key concerns raised by the reviewers, I tend to think that it would not yet be fully convincing for the most critical reviewer. Especially, in terms of soundness of the proposed approach, one reviewer would appear to have key concerns, despite the efforts put in the rebuttal. In view of this, I recommend rejection, but would remain open to discuss this further with the SAC.

**Reviewer Concerns:**

__Regarding validation via non-MCTS rollouts.__ The reviewers were concerned that use of MCTS-based trajectories may yield insufficient comparison, and suggested including some other non-MCTS trajectories. As the rebuttal clarifies, the results also includes LLM-as-a-judge, which is a non-MCTS sampling approach. In conclusion, results from PPRM demonstrate consistent outperformance beyond MCTS rollouts.

__Ablation study.__ Lack of a proper ablation study was raised by some reviewers as a key concern. An ablation study is conducted (and included in Section B.3) with the aim of distinguishing the effects of preference training and new advantage estimator.

__Validation beyond math-related datasets.__ A key concern was that the empirical validation is only done on math-related datasets. The additional experimental results, reported in Section B.4, includes three non-math benchmarks: LeetCode, LiveCodeBench, MMLU-STEM. The reported results shows that policies trained with PPRM achieve state-of-the-art performance on these benchmarks, indicating that its superior performance holds beyond initially considered math-related benchmarks.

__Evaluation on a single backbone.__ Having conducted experiments on a single model was raised as an issue.  The additional experimental results (Section B.5) consider two additional base models: Mistral-MetaMath-7B and DeepSeek-Math-7B-Base. The reported results demonstrate consistent superiority of PPRM over existing methods on these.

__Presentational issues.__ There were some concerns regarding unclarity of motivation in the introductory sections. This issue was addressed well in the rebuttal.

__Insufficient coverage of prior work.__ A reviewer commented that the theory part of the paper overlooks some key studies that investigate bypassing the issues of hard label estimation in PRM. The rebuttal elaborates on how the introduced approach, PPRM, compares with the mentioned references ---they are added in the revision too. In terms of empirical validation, I could verify that the additional experiments demonstrate that PPRM yields superior performance, when Eurus (from the papers cited by the reviewer) is included as a baseline. The rebuttal mentions that PQM is also includes as a baseline, but I am unable to find the corresponding results.

**Reviewer Scores:**

- Reviewer FHML: Since the reviewer's raised concerns are addressed by adding additional experimental results, I would see it likely that the reviewer would be willing to increase the score.
- Reviewer N7uk: I expect the reviewer would maintain the score.
- Reviewer meEZ: Although the key concerns raised by the reviewer are addressed, as far as I can tell, I tend to think that the reviewer would remain negative, especially after taking into account their confidence. I see it possible that they would slightly improve the score, which would still indicate a reject.

---

### Decision · Program_Chairs · 2026-01-26

Reject